# Evolution of Cooperation in Social Dilemmas with Assortative Interactions

**Swami Iyer [1] and Timothy Killingback [2],***

[1]   Department of Computer Science, University of Massachusetts, Boston, MA 02125, USA;
     swami.iyer@gmail.com
[2]   Department of Mathematics, University of Massachusetts, Boston, MA 02125, USA
*    Correspondence: timothy.killingback@umb.edu

**Abstract:** Cooperation in social dilemmas plays a pivotal role in the formation of systems at all levels of complexity, from replicating molecules to multi-cellular organisms to human and animal societies. In spite of its ubiquity, the origin and stability of cooperation pose an evolutionary conundrum, since cooperation, though beneficial to others, is costly to the individual cooperator. Thus natural selection would be expected to favor selfish behavior in which individuals reap the benefits of cooperation without bearing the costs of cooperating themselves. Many proximate mechanisms have been proposed to account for the origin and maintenance of cooperation, including kin selection, direct reciprocity, indirect reciprocity, and evolution in structured populations. Despite the apparent diversity of these approaches they all share a unified underlying logic: namely, each mechanism results in assortative interactions in which individuals using the same strategy interact with a higher probability than they would at random. Here we study the evolution of cooperation in both discrete strategy and continuous strategy social dilemmas with assortative interactions. For the sake of tractability, assortativity is modeled by an individual interacting with another of the same type with probability $r$ and interacting with a random individual in the population with probability $1 - r$, where $r$ is a parameter that characterizes the degree of assortativity in the system. For discrete strategy social dilemmas we use both a generalization of replicator dynamics and individual-based simulations to elucidate the donation, snowdrift, and sculling games with assortative interactions, and determine the analogs of Hamilton's rule, which govern the evolution of cooperation in these games. For continuous strategy social dilemmas we employ both a generalization of deterministic adaptive dynamics and individual-based simulations to study the donation, snowdrift, and tragedy of the commons games, and determine the effect of assortativity on the emergence and stability of cooperation.

**Keywords:** evolutionary game theory; replicator dynamics; adaptive dynamics; prisoners dilemma; hawk-dove game; coordination game; tragedy of the commons

## 1. Introduction

The evolution and stability of cooperative behavior in social dilemmas is a key aspect of the formation of biological systems at multiple levels of complexity, ranging from replicating molecules, at the lower level, to multi-cellular organisms, at the mid level, to human societies, at the high level. Some examples of cooperation in social dilemmas include: formation of early replicating molecules to form larger replicating systems capable of enhanced information encoding [1,2]; integration of the originally autarchic prokaryote precursors of mitochondria and chloroplasts into eukaryotic cells [2]; differential production of replication enzymes in an RNA phage [3]; blood meal donation to roost mates by vampire bats [4]; predator inspection in fish [5]; allogrooming in social mammals [6];

alarm calls in response to danger by mammals and birds [7]; contribution to a wide variety of public goods [8], including, social security, health and welfare programs; restraint in consuming common pool resources [9,10], such as responsible use of fishing stocks, limiting the emission of pollution into the atmosphere, and sharing Internet bandwidth; correct implementation of the Transmission Control Protocol (TCP) so as to avoid congestion in Internet traffic [11]; and file sharing over peer-to-peer networks [12].

Despite the widespread emergence of cooperation in social dilemmas it has proved to be fundamentally challenging to achieve a satisfactory understanding of the origin and maintenance of this phenomenon [2,13–18]. The difficulty in achieving such an understanding is a direct consequence of the nature of a social dilemma itself. A social dilemma may be defined, in classical game theory, as a game which possesses at least one socially inefficient Nash equilibrium [16,19], and in evolutionary game theory, as a game that possesses at least one socially inefficient evolutionary attractor, such as an evolutionary stable strategy [20,21], a stable equilibrium point of the replicator dynamics [22,23], a convergent stable singular strategy of the adaptive dynamics [24–27], or an attracting state in stochastic evolutionary dynamics [28–33]. Since in a social dilemma adopting the strategy at the socially inefficient equilibrium or attractor constitutes defection, while adopting the socially efficient strategy is considered to be cooperation, the nature of the dilemma is that individuals employing strategies corresponding to the socially inefficient attractor will be trapped there by the evolutionary dynamics, despite all individuals being better off if they adopted socially efficient behavior.

As an aside it is interesting to note that many acts of cooperation (when the word cooperation is taken in the literal sense meaning the "process of working together to the same end") may have a direct benefit to the cooperating individual that exceeds the cost (see for example [34]). The emphasis on considering cooperation in social dilemmas in theoretical work is due to this case providing the most challenging theoretical problems, rather than cooperation in social dilemmas being necessarily more common than other forms of cooperation.

A considerable number of different approaches to understanding the evolution of cooperation in social dilemmas have been studied [35]. These include: kin selection [13,36], direct reciprocity [14,37–39], indirect reciprocity [40,41], evolution in network structured populations [19,42–75], and evolution in group structured populations [76,77]. The underlying logic in all these approaches is that they all result in individuals assorting positively, that is, individuals of the same type interact with a greater probability than they would at random. The fundamental role of assortativity in promoting cooperative behavior was already clearly recognized by Hamilton in his work on inclusive fitness [78]. The central position occupied by assortativity in the evolution of cooperation through kin selection was made even more explicit by Grafen in his geometric interpretation of relatedness [79,80]. Grafen's interpretation of relatedness in terms of assortativity is also discussed in [81].

Assortativity is clearly also a key feature of many other proximate mechanisms for promoting the evolution of cooperation. For example, in network structured populations [37,43–64,66–70,72,73] and in group structured populations [76,77], the formation of clusters of cooperators results in preferential interactions between cooperators [50]. The general role played by assortativity in the evolution of cooperation has also been considered recently in [82].

We should emphasize here that many of the most important challenges in understanding the evolution of cooperation center on identifying the different proximate mechanisms that give rise to the assortativity that results in cooperation. However, it is also important to study the effect of assortativity in its own right on the evolution of cooperation. The most significant reason for studying assortativity in its own right is that if it can be shown that positive assortativity leads to some particular outcome in the evolution of cooperation then that provides a rationale for expecting that that outcome will occur for many different proximate mechanisms for obtaining cooperation.

In the present paper we give a detailed and systematic study of the effect of positive assortative interactions, modeled in the manner originally suggested by [79,80], on the evolution of cooperation in a wide variety of both discrete and continuous strategy social dilemmas. In such a system, an individual

interacts with another of the same type with probability $r$ and interacts with a random individual in the population with probability $1 - r$ (where $r$ is a parameter that characterizes the degree of assortativity in the system).

In this paper we take assortativity to be "positive" in the sense that a given individual has a higher probability of interacting with its own type than would be expected at random. The main reason why we focus here on this case is that most mechanisms through which assortativity naturally emerges result in positive assortativity. Perhaps the paradigm of such a situation is evolutionary dynamics in a spatially structured population. In such a population the spatially local nature of the interaction and reproduction processes typically results in the offspring of an individual being localized in the same region of space, and consequently individuals of the same type will often interact with a higher probability than expected from their frequencies in the total population. Thus, evolutionary dynamics in spatially structured populations often results in positive assortativity.

It would be interesting to also consider the possibility of "negative" assortment, in which a given individual would interact with its own type with a lower probability than expected at random. To allow both positive and negative assortativity in a model would require that the degree of assortativity should be dependent on the strategy adopted by the individual, since an individual with a given strategy clearly cannot assort both positively and negatively. In the case of $2 \times 2$ games the generalization to also allow negative assortment would be relatively straight forward. However, for finite strategy games with more than two strategies or for continuous strategy games, the models would be considerably more complicated. For example, in the case of a continuous strategy game, it would be necessary to specify an "assortativity kernel" that would describe precisely the amount of positive or negative assortativity that any given strategy experiences. Despite the possible complications that may arise, studying the effects of negative as well as positive assortment seems to be an interesting area for future research.

It should also be noted that a different and important approach to elucidating the general role played by assortative interactions in promoting the evolution of cooperation focuses on the use of Price's equation [83]. This approach was first developed by Hamilton [78], and has been more recently studied in [84].

We consider here three discrete strategy social dilemmas: the donation game, the snowdrift game, and the sculling game. The donation game is the fundamental exemplar in the prisoner's dilemma class of games, and provides the basic game theory model for altruism [13,17,18]. The snowdrift game is an exemplar of a social dilemma in the hawk-dove class of games [15,18]. While games of hawk-dove type have been extensively studied as models of conflicts and contests [20,21,85], the snowdrift game provides an interesting model for certain types of cooperative behavior that differ from pure altruism [15,17]. The sculling game, is an exemplar of a social dilemma in the coordination class of games. Games in this class have been widely used as models for conventions [28,29,86], but they have typically received little attention as models of cooperation, although interesting exceptions to this trend are [56,70,87,88]. The sculling game, as we define it [19], serves as a model for certain types of cooperative behavior not described by the donation or snowdrift games.

We must emphasize here that there is a very substantial literature on the evolution of cooperation in the prisoner's dilemma with assortative interactions, and we do not claim any great novelty for our results in this case. Equivalent results to those that we have obtained for the discrete donation game can therefore be found in the literature [13,79,80,89–98]. We have discussed the donation game essentially for completeness and to allow comparison with the more novel cases of the snowdrift game and the sculling game. Furthermore, the interesting work [98] considers some closely related issues. The minimum level of assortativity required to allow cooperation to be stably maintained in the prisoner's dilemma, hawk-dove, and stag hunt games is described in terms of the payoff matrix entries in [98], and this leads to a different formulation of some of the results that we obtain here for the discrete strategy games.

Cooperative behavior is often not discrete in nature. This is true of many of the examples of cooperation given above: for instance, when vampire bats share a blood meal with a roost mate [4]. Such situations can be described very naturally using social dilemmas formulated in terms of continuous strategy games. The strategies of individuals in the game represent the level of cooperation, or investment, that they make, and are described by continuous variables. The costs and benefits associated with given investments are represented as continuous functions of the investments. Here we consider three continuous strategy games: the continuous donation game [50,53], in which a cooperative investment made by one individual (the donor or investor) towards another individual (the recipient) benefits the recipient but is costly to the donor; the continuous snowdrift game [26], in which the investment benefits both the donor and the recipient but is also costly to the donor; and the continuous tragedy of the commons game [27], in which the investment—in this context an investment typically represents the level of consumption of a limited common-pool resource, and cooperative behavior correspond to modest levels of consumption—benefits the investor but is also costly to both the investor and the recipient.

For the continuous donation game, just as for the corresponding discrete game, in a well-mixed population cooperation will never evolve: that is, the investments made by individuals in the continuous game will evolve to zero for any cost and benefit functions [50,99]. Various mechanisms have been proposed for the emergence and maintenance of cooperative investments in the game including: spatial [50] and network [53] structure, and reciprocal altruism [99].

For the continuous snowdrift game [26] and continuous tragedy of the commons game [27] in a well-mixed population a variety of different evolutionary outcomes are possible, depending on the nature of the cost and benefit functions. For example, for some cost and benefit functions the evolutionary end state of the population consists of all individuals making the same non-zero investment. However, for other cost and benefit functions high and low investing individuals coexist—an outcome termed the "Tragedy of the Commune" [26]. In the Tragedy of the Commune there is a two-fold social dilemma—not only does evolutionary dynamics result in socially inefficient behavior, but furthermore it forces an unequal outcome in which some individuals make large investments while others invest little or nothing.

Here we consider the effects of positive assortativity on the evolution of cooperation in the continuous donation game, snowdrift game and tragedy of the commons game. We discuss the consequences of assortative interactions both for the level of cooperation that arises and for the emergence of dimorphic evolutionary end states. Related works consider the evolutionary dynamics of continuous strategy games with interaction structure [34,100,101] and the connection to inclusive fitness theory [98]. The interesting work [101] is of particular relevance to our work here in that it considers the effect of relatedness in promoting cooperation in a multi-player version of the continuous snowdrift game. In this work relatedness is introduced in a more general manner to our definition of assortativity: the authors of [101] consider a probability distribution over the number of co-players that are identical-by-descent to a focal individual, and then quantify this assortment distribution based upon the mean and variance of the assortative interactions.

The remainder of this article is organized as follows. In the Models section we formulate the three discrete games in terms of a single cost-to-benefit ratio parameter $\rho$. We analyze the games using the framework of replicator dynamics generalized to include the degree of assortativity $r$, and derive mathematical relations (analogs of Hamilton's rule) involving parameters $\rho$ and $r$ that must be satisfied for the evolution of complete cooperation. We also formulate and analyze the three continuous strategy games using the framework of adaptive dynamics generalized to allow for assortative interactions, and derive conditions that determine the effect of assortativity on the stability of cooperation. In the Results section we describe the results of studying the evolution of cooperation in these discrete and continuous strategy games with assortative interactions using an individual-based model, and we compare the results obtained from these simulations with those obtained analytically from the assortative generalizations of replicator dynamics and adaptive dynamics. Finally, in the

Discussion section we conclude the article with a discussion of the significance of our results and with some suggestions for further inquiry.

## 2. Models

### 2.1. Discrete Games

#### 2.1.1. Replicator Dynamics with Assortative Interactions

In general, consider a 2-player game with $m$ pure strategies $\sigma_1, \ldots, \sigma_m$ and strategy space $\mathcal{S} = \{\sigma_1, \ldots, \sigma_m\}$. Let $\mathcal{P}$ be a large population of individuals, each of which uses a strategy from $\mathcal{S}$. We define assortative interactions as follows. An assortative interaction among individuals is an interaction that occurs preferentially among individuals of the same type, i.e., such an interaction occurs among individuals of the same type with a greater probability than would occur through random interactions. In the context of game theory, this means that individuals using the same strategy interact with a probability greater than that which would occur with random interactions. The most direct and convenient way to introduce assortative interactions in a population is to specify a parameter $r \in [0,1]$, called the degree of assortativity or simply the assortativity, which is defined as follows: with probability $r$, an individual interacts with another individual of its own type, and with probability $1 - r$, the individual interacts with a randomly chosen individual from the population [79,80]. Thus, we say that the population $\mathcal{P}$ is assortatively-interacting or assortatively-mixed, with assortativity $r$, if an individual in $\mathcal{P}$ interacts with another individual in $\mathcal{P}$ with the same strategy with probability $r$ and with a randomly picked individual in $\mathcal{P}$ with probability $1 - r$.

Let the payoff to strategy $\sigma_i$ against $\sigma_j$ be denoted by $\pi(\sigma_i, \sigma_j)$. In biological evolution $\pi(\sigma_i, \sigma_j)$ represents the change in the Darwinian fitness (that is, the change in the expected number of offspring) of an individual using strategy $\sigma_i$ in an interaction with an individual using $\sigma_j$. In social evolution through imitation dynamics $\pi(\sigma_i, \sigma_j)$ measures the utility obtained by an individual using strategy $\sigma_i$ in an interaction with an individual using $\sigma_j$. At time $t$, let $N(t)$ be the total size of the population $\mathcal{P}$, and let $n_i(t)$ be the number of individuals in $\mathcal{P}$ using strategy $\sigma_i$. The frequency of strategy $\sigma_i$ is defined to be $p_i(t) = \frac{n_i(t)}{N(t)}$. If $\mathbf{p}(t) = (p_1(t), \ldots, p_m(t))$ denotes the vector of frequencies, then the state of the population at time $t$ is given by $\mathbf{p}(t) \in \Delta_m$, where $\Delta_m = \{\mathbf{p}(t) \in \mathbb{R}^m_+ : \sum_{i=1}^m p_i(t) = 1\}$ is the $m$-simplex.

We now consider how the frequencies of the different strategies $\sigma_i$ change with time in an assortatively-interacting population $\mathcal{P}$, with assortativity $r$, due to natural selection. The fitness $f_i^r$ of $\sigma_i$ is the average payoff of $\sigma_i$, which, from the definition of assortativity, is given by

$$f_i^r = r\pi(\sigma_i, \sigma_i) + (1 - r) \sum_{j=1}^m p_j \pi(\sigma_i, \sigma_j). \tag{1}$$

The dynamics of $\mathbf{p}(t)$ on $\Delta_m$ in an assortatively-interacting population can be determined (in analogy with the non-assortative case [17,18,22,23,102]) as follows. The growth rate of the number of individuals $n_i$ using strategy $\sigma_i$ is $\dot{n}_i = n_i f_i^r$. Therefore,

$$\begin{aligned} \dot{p}_i = \frac{d}{dt}\left(\frac{n_i}{N}\right) &= \frac{\dot{n}_i N - n_i \sum_j \dot{n}_j}{N^2} \\ &= p_i f_i^r - p_i \sum_j p_j f_j^r \\ &= p_i(f_i^r - \bar{f}^r), \end{aligned} \tag{2}$$

where $\bar{f}^r = \sum_j p_j f_j^r$ is the mean fitness of the strategies in the population. This equation is the analog of the standard replicator equation for an assortatively-interacting population. We shall refer to this equation as the assortative replicator equation with assortativity $r$, or the $r$-replicator equation for

short. The $r$-replicator equation is the natural generalization of the standard replicator equation when assortative interactions are included in the manner proposed by [79,80]. The $r$-replicator equation is equivalent to the standard (non-assortative) replicator equation under a transformation of the payoffs. If $A = (a_{ij})$, where $a_{ij} = \pi(\sigma_i, \sigma_j)$, is the payoff matrix, then it is elementary to verify that the replicator equation with transformed payoff matrix $\hat{A} = rB + (1 - r)A$, where $B = (b_{ij})$ is the matrix such that $b_{ij} = a_{ii}$, is equivalent to the $r$-replicator equation. Notwithstanding this equivalence, however, we feel that the best way to think about the replicator equation when there are assortative interactions is to consider the assortativity as directly affecting how the fitness is defined, as we have done here, since the change in the fitness is clear and natural, whereas the necessary transformation in the payoffs is less clear.

Grafen did not formulate the $r$-replicator equation in his work, since it was unnecessary for his aim of studying evolutionary stable strategies with assortative interactions. Equations similar to the $r$-replicator equation have been considered for assortativity that depends on the frequencies of the strategies by [91] and for different population structures (including those that include relatedness) by [92].

For $r = 0$, the $r$-replicator equation reduces to the standard replicator equation for a well-mixed population [17,18,22,23,102]:

$$\dot{p}_i = p_i(f_i - \bar{f}), \tag{3}$$

where $\bar{f} = \sum_j p_j f_j$ is the mean fitness of the population.

Consider now the case of a symmetric $2 \times 2$ game, with strategies denoted by $C$ and $D$, and with payoff matrix $\pi$ given by

$$\pi = \begin{array}{c} \\ C \\ D \end{array} \begin{array}{c} C \quad D \\ \begin{bmatrix} \alpha & \beta \\ \gamma & \delta \end{bmatrix} \end{array}, \tag{4}$$

where $\alpha, \beta, \gamma, \delta \in \mathbb{R}$. Let $p$ denote the frequency of strategy $C$ in the population, and thus $1 - p$ is the frequency of strategy $D$.

Now consider a population of assortatively-interacting individuals who are playing this $2 \times 2$ game. Let $r$ denote the degree of assortativity in the population. The fitnesses $f_C^r$ and $f_D^r$ of the strategies $C$ and $D$, with assortativity, are given by

$$\begin{aligned} f_C^r &= r\pi(C, C) + (1 - r)[p\pi(C, C) + (1 - p)\pi(C, D)] \\ &= r\alpha + (1 - r)[p\alpha + (1 - p)\beta] \text{ and} \end{aligned} \tag{5}$$

$$\begin{aligned} f_D^r &= r\pi(D, D) + (1 - r)[p\pi(D, C) + (1 - p)\pi(D, D)] \\ &= r\delta + (1 - r)[p\gamma + (1 - p)\delta], \end{aligned} \tag{6}$$

and the average fitness of the population is $\bar{f}^r = pf_C^r + (1 - p)f_D^r$. The evolutionary dynamics of the population, with assortativity, is thus given by the $r$-replicator equation

$$\begin{aligned} \dot{p} &= p(f_C^r - \bar{f}^r) \\ &= p(1 - p)(f_C^r - f_D^r) \\ &= p(1 - p)\{r(\alpha - \delta) + (1 - r)[p(\alpha - \gamma) + (1 - p)(\beta - \delta)]\}. \end{aligned} \tag{7}$$

We note that for $r = 0$, this equation reduces to the standard replicator equation for a symmetric $2 \times 2$ game [17,18,22,23,102]):

$$
\begin{aligned}
\dot{p} &= p(f_C - \bar{f}) \\
&= p(1 - p)(f_C - f_D) \\
&= p(1 - p)[p(\alpha - \beta - \gamma + \delta) + \beta - \delta],
\end{aligned}
\tag{8}
$$

where $f_C = p\pi(C, C) + (1 - p)\pi(C, D)$ and $f_D = p\pi(D, C) + (1 - p)\pi(D, D)$ are the fitnesses of the strategies $C$ and $D$, respectively, and $\bar{f} = pf_C + (1 - p)f_D$ is the mean fitness of the population.

Here, for simplicity, in the rest of this paper when we discuss discrete strategy games we shall restrict our attention to two-strategy games—however, our analysis of evolutionary dynamics in assortatively-mixed populations can be extended to games with any number of strategies, and this may represent an interesting topic for future study.

### 2.1.2. Donation Game

The donation game is the fundamental exemplar in the prisoner's dilemma class of games, and provides the basic game theory model for altruism [13,17,18]. We must emphasize here that there is significant literature on the evolution of cooperation in the prisoner's dilemma with assortative interactions, and we do not claim any particular originality for our results in this case. We discuss the donation game here essentially for completeness and to allow comparison with the more novel cases of the snowdrift game and the sculling game that follows. Equivalent results to those that we obtain here for the discrete donation game can therefore be found in the literature [13,79,80,89–98].

Consider the situation of two individuals, John and Bill, who donate blood to each other, as described in [19]. Suppose that the act of donating blood to someone incurs a cost $c$ to the donor but confers a benefit $b$ to the recipient, where $b, c \in \mathbb{R}_+$ and $b > c$. If we consider donation as the cooperative strategy $C$ and non-donation as the defective strategy $D$, then the payoff matrix for the donation game is given by [19]:

$$
\pi = \begin{array}{c} \\ C \\ D \end{array} \overset{\begin{array}{cc} C & D \end{array}}{\begin{bmatrix} 1 - \rho & -\rho \\ 1 & 0 \end{bmatrix}},
\tag{9}
$$

where $\rho \in (0, 1)$ is the cost-to-benefit ratio $\rho = \frac{c}{b}$. It follows directly from the rank ordering of the elements of the payoff matrix that the game is in the prisoner's dilemma class of symmetric $2 \times 2$ games.

The standard replicator Equation (8) gives the evolutionary dynamics for the game without assortative interactions to be $\dot{p} = -p(1 - p)\rho$. The equilibrium points for the dynamics are $\hat{p} = 0$ and $\hat{p} = 1$, the former being asymptotically stable, while the latter is unstable. Thus, a well-mixed population of individuals playing the donation game, starting from an initial frequency $p_0 \in (0, 1)$ of cooperators, will evolve towards the all-defector ($\hat{p} = 0$) equilibrium state [19].

Consider now an assortatively-interacting population playing the donation game with the payoff matrix given by (9). Let the degree of assortativity be $r$. It follows directly from the $r$-replicator Equation (7) that the evolutionary dynamics with assortativity for the population is given by

$$
\dot{p} = p(1 - p)(r - \rho).
\tag{10}
$$

Figure 1 shows the phase line diagrams and the bifurcation diagram for the game. The system has two equilibria, $\hat{p} = 0$ and $\hat{p} = 1$. For $r < \rho$, the equilibrium point $\hat{p} = 0$ is asymptotically stable, while the equilibrium point $\hat{p} = 1$ is unstable. Thus, a population, starting from an initial frequency $p_0 \in (0, 1)$ of cooperators, will evolve towards the stable equilibrium $\hat{p} = 0$, representing a state in which all individuals defect. However, a bifurcation occurs at $r_c = \rho$, reversing the stability

of the two equilibrium points. Hence, if $r > \rho$ a population starting with any positive frequency of cooperators will evolve towards the stable equilibrium $\hat{p} = 1$, representing a state in which all individuals cooperate. A similar inequality is arrived at in [92] for the prisoner's dilemma game that has constant gains from switching (that is, in which fitness effects are additive), with the constant $r$ representing the population structure and $\rho$ the ratio of total costs to total benefits (not including self).

Thus, as is essentially well-known, assortative interactions provide a mechanism for promoting cooperative behavior in the donation game [13,79,80,89–98]. It should be noted that the condition for cooperation to evolve through assortativity in the donation game, $r > \rho$, is formally identical to Hamilton's inequality, which governs the evolution of altruism through kin selection [13]. This correspondence is completely natural given Grafen's geometric interpretation of relatedness in terms of assortativity [79,80].

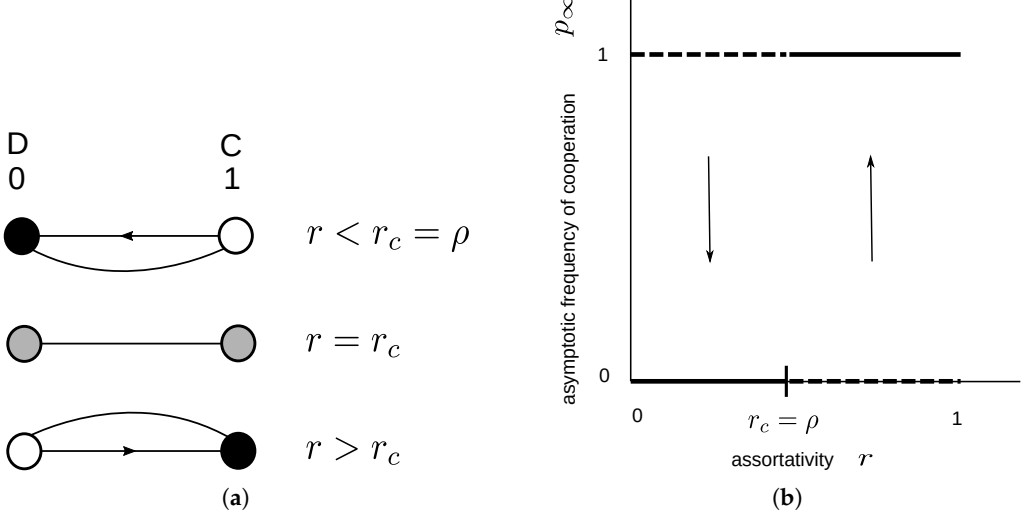

**Figure 1.** Phase line diagrams and the bifurcation diagram for the donation game with assortative interactions. (**a**) In the phase line diagrams closed circles represent stable equilibrium points, open circles represent unstable equilibrium points, and the curved line connecting equilibrium points indicates the graph of the function on the right-hand side of the *r*-replicator equation. (**b**) In the bifurcation diagram solid lines represent stable equilibrium points, dashed lines represent unstable equilibrium points, and arrows indicate the direction of evolutionary change.

### 2.1.3. Snowdrift Game

The snowdrift game is an interesting exemplar of a social dilemma in the hawk-dove class of games, and provides a model for certain types of cooperative behavior that differ from pure altruism [15,17,18]. Consider the situation described in [19] of two individuals, John and Bill, who are stuck in a car on their way home because the road is blocked by a snowdrift. Let the benefit of getting home be $b$ and the cost of clearing the snow be $c$, where $b, c \in \mathbb{R}_+$ and $b > c$. If we consider shoveling as the cooperative strategy $C$ and non-shoveling as the defective strategy $D$, then the payoff matrix for the snowdrift game is given by [19]:

$$\pi = \begin{array}{c} \\ C \\ D \end{array} \overset{\begin{array}{cc} C & D \end{array}}{\begin{bmatrix} 1 - \frac{\rho}{2} & 1 - \rho \\ 1 & 0 \end{bmatrix}}, \tag{11}$$

where $\rho \in (0,1)$ is the cost-to-benefit ratio $\rho = \frac{c}{b}$. It follows immediately from the rank ordering of the elements of the payoff matrix that the snowdrift game is in the hawk-dove class of symmetric $2 \times 2$ games.

In the absence of assortativity, the standard replicator Equation (8) gives the evolutionary dynamics for the game to be $\dot{p} = p(1-p)[p(\frac{\rho}{2}-1)+1-\rho]$. The equilibrium points for the dynamics are $\hat{p} = 0$ and $\hat{p} = 1$, and $p^{\star} = \frac{1-\rho}{1-\frac{\rho}{2}}$. The latter internal equilibrium is asymptotically stable, while the former two boundary equilibria are unstable. Thus, a population of individuals playing the snowdrift game, starting from an initial fraction $p_0 \in (0,1)$ of cooperators, will evolve towards the internal equilibrium state ($p^{\star} = \frac{1-\rho}{1-\frac{\rho}{2}}$) in which cooperators and defectors coexist [19].

Consider a population of assortatively-interacting individuals playing the snowdrift game with the payoff matrix given by (11). Let the degree of assortativity be $r$. The $r$-replicator Equation (7) gives the evolutionary dynamics with assortativity for the population to be

$$\dot{p} = p(1-p)\left[p(1-r)\left(\frac{\rho}{2}-1\right)+\frac{r\rho}{2}-\rho+1\right]. \tag{12}$$

Figure 2 shows the phase line diagrams and the bifurcation diagram for the game. The system has three equilibria: $\hat{p} = 0$ and $\hat{p} = 1$ at the boundaries, and a possible internal equilibrium

$$p^{\star} = \frac{\frac{r\rho}{2}+1-\rho}{(1-r)\left(1-\frac{\rho}{2}\right)}. \tag{13}$$

It is easy to verify that if $r < r_c = \frac{\rho}{2}$ then the boundary equilibria are unstable, while there exists an internal equilibrium $p^{\star}$ which is stable. Thus, if $r < r_c$ then a population starting with an initial frequency $p_0 \in (0,1)$ of cooperating individuals will evolve to a state of coexistence determined by the internal equilibrium, in which a fraction $p^{\star}$ of the population will be cooperators and the remainder will be defectors. A bifurcation occurs at $r = r_c$, in which the equilibrium $p^{\star}$ passes through the boundary equilibrium $\hat{p} = 1$, resulting in a change of stability for the boundary equilibria. For $r > r_c$ there no longer exists any internal equilibrium, while the boundary equilibrium $\hat{p} = 0$ becomes unstable and the equilibrium $\hat{p} = 1$ becomes globally asymptotically stable. Therefore, for $r > r_c$ a population starting with any positive initial frequency $p_0$ of cooperators will evolve to the completely cooperative state determined by the equilibrium $\hat{p} = 1$.

For $r < r_c = \frac{\rho}{2}$, in which case there is a stable internal equilibrium $p^{\star}$, it follows from $\frac{\partial p^{\star}}{\partial r} = \frac{1-\rho(1-\frac{\rho}{4})}{[(1-r)(1-\frac{\rho}{2})]^2} > 0$, that increasing the assortativity $r$ has the effect of increasing the frequency of cooperators at equilibrium. Similarly, in this case, since $\frac{\partial p^{\star}}{\partial \rho} = \frac{r-\frac{r^2}{2}-\frac{1}{2}}{[(1-r)(1-\frac{\rho}{2})]^2} < 0$, increasing the cost-to-benefit ratio $\rho$ decreases the frequency of cooperators at equilibrium. Furthermore, for $r > r_c = \frac{\rho}{2}$, the population is completely cooperative. Thus, for the snowdrift game, cooperative behavior is promoted in an assortatively-interacting population, and the condition $r > \frac{\rho}{2}$ governing the transition to complete cooperation is an analog of Hamilton's inequality for the donation game. In related work, the minimal level of assortativity required for cooperation to be stably maintained in hawk-dove games is studied in [98].

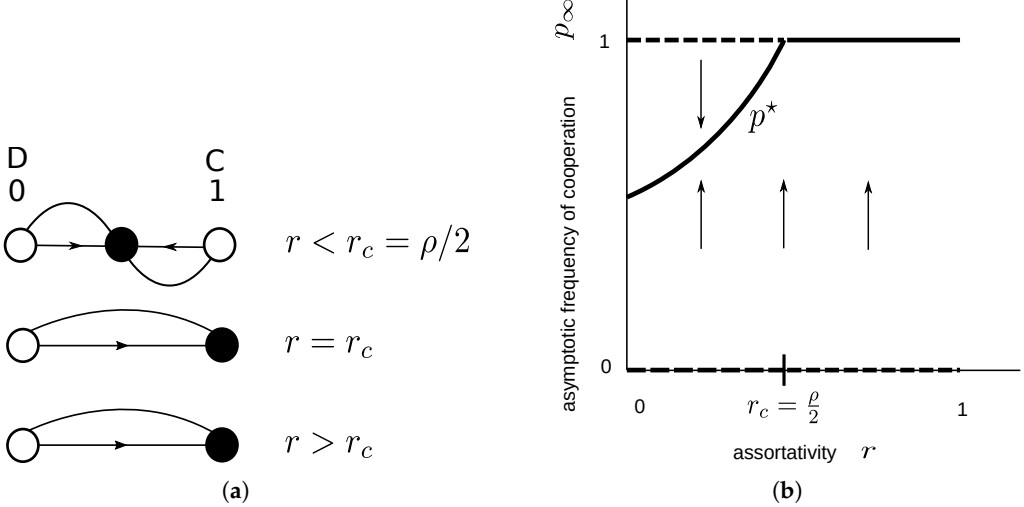

**Figure 2.** Phase line diagrams and the bifurcation diagram for the snowdrift game with assortative interactions. (**a**) In the phase line diagrams closed circles represent stable equilibrium points, open circles represent unstable equilibrium points, and the curved line connecting equilibrium points indicates the graph of the function on the right-hand side of the *r*-replicator equation. (**b**) In the bifurcation diagram solid lines represent stable equilibrium points, dashed lines represent unstable equilibrium points, and arrows indicate the direction of evolutionary change.

### 2.1.4. Sculling Game

Games in the coordination class have typically received little attention as models of cooperation, although for notable exceptions to this tendency see [56,70,87,88]. Here we consider the sculling game [19] as an exemplar of a social dilemma in the coordination class of games, with the aim of using it as a model for certain types of cooperative behavior not described by the donation or snowdrift games. Suppose individuals John and Bill are rowing in a double scull to get to their destination, as described in [19]. Let reaching the destination have a value of $\frac{b}{2}$ to both players, where $b \in \mathbb{R}_+$, and let the cost of rowing be $c \in \mathbb{R}_+$ to the rower. If we treat rowing as the cooperative strategy $C$ and non-rowing as the defective strategy $D$, then the payoff matrix for the sculling game is given by [19].

$$
\pi = \begin{array}{c} \\ C \\ D \end{array} \begin{array}{cc} C & D \\ \left[ \begin{array}{cc} 2-\rho & \frac{1}{2}-\rho \\ \frac{1}{2} & 0 \end{array} \right], \end{array}
\tag{14}
$$

where $\rho \in (\frac{1}{2}, \frac{3}{2})$ is the cost-to-benefit ratio $\rho = \frac{c}{b}$. It follows directly from the rank ordering of the elements of the payoff matrix that the sculling game is in the coordination class of symmetric $2 \times 2$ games.

Equation (8) gives the evolutionary dynamics for the game to be $\dot{p} = p(1-p)[p + \frac{1}{2} - \rho]$. The equilibrium points for the dynamics are $\hat{p} = 0$, $\hat{p} = 1$, and $p^\star = \rho - \frac{1}{2}$. The former two boundary equilibria are asymptotically stable, while the latter internal equilibrium is unstable. Thus, a population of individuals playing the sculling game, starting from an initial fraction $p_0 \in (0,1) \backslash \{\rho - \frac{1}{2}\}$ of cooperators, will evolve towards the all-defector ($\hat{p} = 0$) state if $p_0 < \rho - \frac{1}{2}$, and towards the all-cooperator ($\hat{p} = 1$) state if $p_0 > \rho - \frac{1}{2}$. We note that the equilibrium $(C, C)$ is payoff-dominant over $(D, D)$, for all $\rho$. Moreover, $(C, C)$ is risk-dominant for $\rho \in (\frac{1}{2}, 1)$, while $(D, D)$ is risk-dominant for $\rho \in (1, \frac{3}{2})$. A coordination game in which one of the two equilibria is payoff dominant while the other is risk dominant is a stag hunt game [88]. Thus, for $\rho \in (1, \frac{3}{2})$ the sculling game is an exemplar of the stag hunt game [19].

Consider a population of assortatively-interacting individuals playing the sculling game with the payoff matrix (14). Let the degree of assortativity be $r$. The $r$-replicator Equation (7) gives that the evolutionary dynamics with assortativity for the population is given by

$$\dot{p} = p(1-p)\left[p(1-r) + \frac{3r}{2} - \rho + \frac{1}{2}\right]. \tag{15}$$

Figure 3 shows the phase line diagrams and the bifurcation diagram for the game. The system has three equilibria: $\hat{p} = 0$ and $\hat{p} = 1$ at the boundaries, and a possible internal equilibrium

$$p^{\star} = \frac{-\frac{3r}{2} + \rho - \frac{1}{2}}{1 - r}. \tag{16}$$

It is straightforward to verify that if $r < r_c = \frac{2\rho}{3} - \frac{1}{3}$ then the boundary equilibria $\hat{p} = 0$ and $\hat{p} = 1$ are stable, and there exists an unstable interior equilibrium $p^{\star}$. Thus, if $r < r_c = \frac{2\rho}{3} - \frac{1}{3}$ then a population starting with an initial frequency $p_0 \neq p^{\star}$ of cooperators will evolve to the all-defector state $\hat{p} = 0$ if $p_0 < p^{\star}$ and to the all-cooperator state $\hat{p} = 1$ if $p_0 > p^{\star}$. It follows from this that, for $r < r_c$, if $r < \hat{r} = \frac{\rho - \frac{1}{2} - p_0}{\frac{3}{2} - p_0}$ then such a population evolves to the all-defector equilibrium $\hat{p} = 0$, while if $r > \hat{r}$ then the population evolves to the all-cooperator equilibrium $\hat{p} = 1$. For example, if a population starts with an initial frequency of cooperators $p_0 = \frac{1}{2}$, then $\hat{r} = \rho - 1$, and the population will evolve to all-defection if $r < \rho - 1$ and to all-cooperation if $r > \rho - 1$. A bifurcation occurs at $r = r_c$, in which the equilibrium $p^{\star}$ passes through the boundary equilibrium $\hat{p} = 0$, resulting in a change of stability for the boundary equilibria. Consequently, for $r > r_c$ there does not exist any internal equilibrium, the boundary equilibrium $\hat{p} = 0$ is unstable, and the equilibrium $\hat{p} = 1$ is globally asymptotically stable. Thus, if $r > r_c$ then a population with any positive initial frequency $p_0$ of cooperators will evolve to the completely cooperative state $\hat{p} = 1$.

For $r < r_c = \frac{2\rho}{3} - \frac{1}{3}$, in which case there is an unstable internal equilibrium $p^{\star}$, it follows from $\frac{\partial p^{\star}}{\partial r} = \frac{\rho - 2}{(1-r)^2} < 0$ that increasing the assortativity $r$ has the effect of increasing the basin of attraction around $\hat{p} = 1$, thus decreasing the value of the initial frequency $p_0$ of cooperators that is needed for the population to evolve to the all-cooperator state. Similarly, in this case, since $\frac{\partial p^{\star}}{\partial \rho} = \frac{1}{1-r} > 0$, increasing the cost-to-benefit ratio $\rho$ has the effect of increasing the basin of attraction around $\hat{p} = 0$, thus increasing the value of the initial frequency $p_0$ of cooperators that is needed for the population to evolve to the all-cooperator state. Furthermore, for $r > r_c = \frac{2\rho}{3} - \frac{1}{3}$, the population becomes completely cooperative. Thus, for the sculling game, cooperative behavior is promoted in an assortatively-interacting population, and the condition $r > \frac{2\rho}{3} - \frac{1}{3}$ governing the transition to complete cooperation is an analog of Hamilton's inequality for the donation game. In related work, the effect of assortativity on the maintenance of cooperation in stag hunt games is considered in [98].

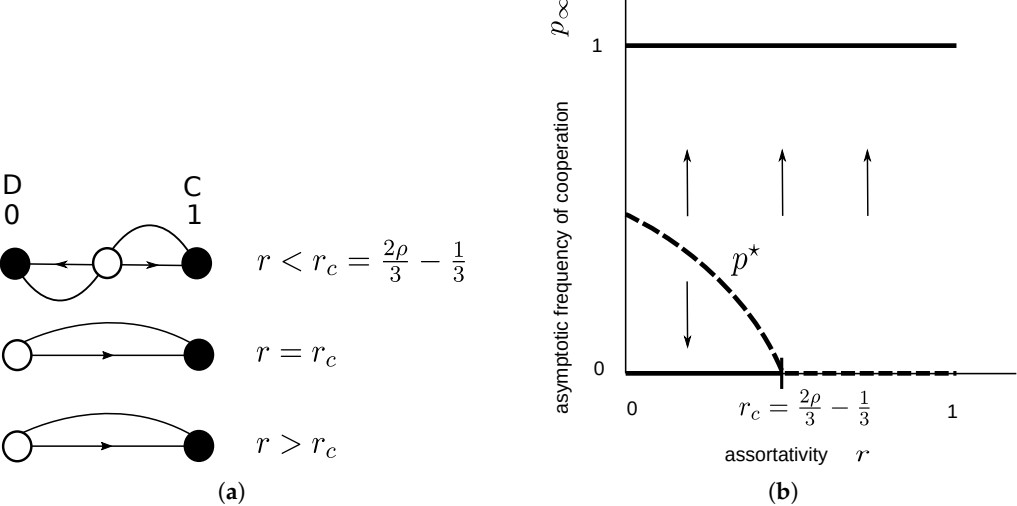

**Figure 3.** Phase line diagrams and the bifurcation diagram for the sculling game with assortative interactions. (**a**) In the phase line diagrams closed circles represent stable equilibrium points, open circles represent unstable equilibrium points, and the curved line connecting equilibrium points indicates the graph of the function on the right-hand side of the *r*-replicator equation. (**b**) In the bifurcation diagram solid lines represent stable equilibrium points, dashed lines represent unstable equilibrium points, and arrows indicate the direction of evolutionary change.

### 2.2. Continuous Games

#### 2.2.1. Adaptive Dynamics with Assortative Interactions

Let us now consider the evolutionary dynamics of continuous strategy social dilemmas with assortative interactions, which we model using continuous strategy games in assortatively-interacting populations. We quantify the probability that an individual interacts with another individual of its own type (i.e., in the context of games, the probability that an individual using a given strategy interacts with another individual using the same strategy) by the parameter $r \in [0, 1]$, the degree of assortativity, in exactly the same way as discussed above for the case of discrete games. Since the exact equivalent of replicator dynamics for continuous strategy games results in an infinite dimensional dynamical system which is difficult to study analytically, here we will adopt the standard approach of using deterministic, monomorphic, adaptive dynamics to study the evolutionary dynamics of the continuous strategy games [17,23–27,103–105]. We will study the evolutionary behavior of continuous games in assortatively-mixed populations analytically using a generalization of standard deterministic adaptive dynamics that accounts for the assortative interactions.

Consider a large assortatively-mixed population of individuals, with assortativity $r$, playing a continuous game. We assume that the interactions are pairwise, with $\pi(x, y)$ denoting the payoff to an $x$-strategist when interacting with a $y$-strategist. Let the population be initially monomorphic with all individuals using the same resident strategy $x$. Now consider a small fraction $\xi$ of mutant individuals in the population playing strategy $y$. The fitnesses $f_x^r$ and $f_y^r$ of the $x$ and $y$ strategies are given by

$$f_x^r = r\pi(x, x) + (1 - r)[(1 - \xi)\pi(x, x) + \xi\pi(x, y)] \text{ and}$$
$$f_y^r = r\pi(y, y) + (1 - r)[(1 - \xi)\pi(y, x) + \xi\pi(y, y)]. \tag{17}$$

The invasion fitness $\phi_x^r(y)$ of the mutant strategy $y$ in the resident strategy $x$ is the per capita growth rate of $y$ when rare, and thus is given by $\phi_x^r(y) = \lim_{\xi \to 0} \frac{\dot{\xi}}{\xi}$. It follows from the $r$-replicator equation $\dot{\xi} = \xi(1 - \xi)(f_y^r - f_x^r)$ that

$$
\begin{aligned}
\phi_x^r(y) &= \lim_{\xi \to 0} \frac{\dot{\xi}}{\xi} \\
&= r\pi(y, y) + (1 - r)\pi(y, x) - \pi(x, x).
\end{aligned} \tag{18}
$$

We note that when $r = 0$, the above equation reduces to

$$
\phi_x^0(y) = \pi(y, x) - \pi(x, x), \tag{19}
$$

which is the standard equation for the invasion fitness of a continuous strategy game in a well-mixed population [26].

The adaptive dynamics of a continuous strategy game in an assortatively-interacting population is determined by the invasion fitness $\phi_x^r(y)$ [17,23–26,104]. The evolution of the strategy $x$ is governed by the selection gradient $D(x) = \left. \frac{\partial \phi_x^r}{\partial y} \right|_{y=x}$, and the adaptive dynamics of $x$ is determined by the differential equation $\dot{x} = mD(x)$, where $m$ depends on the population size and on the mutational process at work [106]. For a constant population size, $m$ simply scales the time variable, and thus we can set $m = 1$ without any loss of generality. Equilibrium points of the adaptive dynamics are called singular strategies and are solutions of $D(x^\star) = 0$. If no such solutions exist, then the strategy $x$ monotonically increases or decreases under evolution, depending on the sign of $D(x)$. If $x^\star$ exists, it is convergent stable and, hence an attractor for the adaptive dynamics, if $\left. \frac{dD}{dx} \right|_{x=x^\star} < 0$. If this equality is reversed, $x^\star$ is a repeller. Initially, the population will converge to a convergent stable singular point $x^\star$, but its subsequent evolutionary fate depends on whether $x^\star$ is a maximum or minimum of the invasion fitness $\phi_x^r(y)$. If $x^\star$ is a maximum, i.e., if $\left. \frac{\partial^2 \phi_{x^\star}^r}{\partial y^2} \right|_{y=x^\star} < 0$, then $x^\star$ is an evolutionarily stable strategy (ESS), representing an evolutionary end state in which all individuals adopt strategy $x^\star$. If, however, $\left. \frac{\partial^2 \phi_{x^\star}^r}{\partial y^2} \right|_{y=x^\star} > 0$, then a population of $x^\star$-strategists can be invaded by mutant strategies on either side of $x^\star$. In this case the population undergoes evolutionary branching and splits into two distinct and diverging clusters of strategies.

### 2.2.2. Continuous Donation Game

Consider the situation of two individuals, John and Bill, who donate blood to each other, as described in the discrete donation game formulated in [19]. Let us assume now that the amount of blood donated by each individual can vary. Interactions among pairs of individuals in which the cooperative investment (or donation) is continuously variable, and that made by one individual (the donor) benefits the other individual (the recipient) but is costly to the donor, can be described using the continuous donation (CD) game (also referred to as the continuous prisoner's dilemma game) [50,53,99].

The CD game involves the interaction between two individuals, making investments $x$ and $y$, respectively, where $x, y \in \mathbb{R}_+$. An investment $x$ has the following effects: the payoff of the investor (donor) is reduced by $C(x)$, where $C$ is a function that specifies the cost of making the investment, and the payoff of the beneficiary (recipient) is increased by $B(x)$, where $B$ is a function that specifies the benefit resulting from the investment. Therefore, the payoff to an $x$-investor interacting with a $y$-investor is given by the relation

$$
\pi(x, y) = B(y) - C(x). \tag{20}
$$

We shall assume here that there is an upper limit $x_m$ on the possible level of investment. We also assume that the cost and benefit functions are smooth and monotonically increasing,

satisfy $C(0) = B(0) = 0$, and $B(x) > C(x)$, for $x \in (0, x_c)$, where $0 < x_c < x_m$. This latter assumption is a necessary condition for cooperation to evolve: if it does not hold, then if every individual invests $x > 0$, then each receives a lower payoff than if each had invested 0.

In a well-mixed population (i.e., without assortativity) the adaptive dynamics of the investment $x$ is given by $\dot{x} = -C'(x)$, and thus since $C'(x) > 0$, for all $x$, it follows that $x$ evolves to 0, the complete defection state. Hence, the CD game is a social dilemma: evolutionary dynamics results in all individuals making zero investment, with each individual consequently obtaining zero payoff, however, if each individual invested $x \in (0, x_c)$, then all individuals would receive a payoff $\pi(x, x) = B(x) - C(x) > 0$.

For an assortatively-mixed population of individuals with assortativity $r$, from Equations (18) and (20), we can write the invasion fitness $\phi_x^r(y)$ as

$$\phi_x^r(y) = r[B(y) - B(x)] - C(y) + C(x). \tag{21}$$

The adaptive dynamics of the investment $x$ is governed by

$$\begin{aligned}
\dot{x} &= D(x) \\
&= \left. \frac{\partial \phi_x^r(y)}{\partial y} \right|_{y=x} \\
&= rB'(x) - C'(x). 
\end{aligned} \tag{22}$$

We note here that under quite general conditions the final evolutionary outcome of the CD game in an assortatively-interacting population is completely determined by the analog of Hamilton's inequality described above. Let us assume first that there is no singular strategy $x^\star \in (0, x_m)$. In this case, if $r > \rho = \frac{C'(0)}{B'(0)}$ then it follows from the continuity of $D(x)$ that $D(x) > 0$, for all $x \in (0, x_m)$, and thus $x$ will evolve to the maximally cooperative state $x = x_m$. If, on the other hand, $r < \rho = \frac{C'(0)}{B'(0)}$ then $x$ will evolve to the totally defective state $x = 0$. Let us now assume that there exists a unique singular strategy $x^\star \in (0, x_m)$, which is non-degenerate (i.e., $D'(x^\star) \neq 0$). In this case, if $r > \rho = \frac{C'(0)}{B'(0)}$ then it follows from the continuity of $D(x)$ that $D'(x^\star) < 0$, and thus $x^\star$ is convergent stable, and as shown above $x^\star$ is therefore also an ESS. Consequently, the population will evolve to a final state in which all individuals are investing $x^\star$. If, however, $r < \rho = \frac{C'(0)}{B'(0)}$ then $D'(x^\star) > 0$, and hence $x^\star$ is a repeller. In this latter case, if the initial state of the population $x_0 < x^\star$ then the population will evolve to the total defection state $x = 0$, while if $x_0 > x^\star$ then the population will evolve to the maximally cooperative state $x = x_m$.

Linear Cost and Benefit Functions

Suppose the cost and benefit functions are linear functions of the investment $x$, i.e., suppose $C(x) = cx$ and $B(x) = bx$, where $b > c$. Linear cost and benefit functions are interesting because they arise as approximations to more general cost and benefit functions.

From Equation (22), $\dot{x} = D(x) = rb - c$. Since $D(x)$ is constant, there are no singular strategies for the game, and the evolutionary fate of a mutant strategy is determined solely by the sign of $D(x)$. An initially monomorphic population in which every individual invests an amount $x_0 \in [0, x_m]$, will evolve to the maximally cooperative state in which all individuals invest $x_m$ if $r > \rho = \frac{C'(0)}{B'(0)} = \frac{c}{b}$. If on the other hand the inequality is reversed, then the population will evolve to the state in which all individuals defect by making zero investment. Note that in this case the condition that governs the evolution of cooperation in the CD game is formally identical to the classical Hamilton's rule [107].

Convex Cost and Concave Benefit Functions

Now we consider the case in which the cost function is convex (i.e., $C''(x) > 0$) and the benefit function is concave ($B''(x) < 0$). There is good evidence that in many situations the benefit function exhibits diminishing returns for sufficiently large levels of investment [108–110], and the cost is often well described by a convex quadratic function [111]. Thus, as an illustrative example we take the cost and benefit functions to be quadratic functions of the investment $x$: $C(x) = c_1 x^2$ and $B(x) = -b_2 x^2 + b_1 x$, where $c_1, b_1, b_2, > 0$.

The evolution of cooperation is determined by the analog of Hamilton's inequality given above: namely, cooperation will evolve if $r > \rho = \frac{C'(0)}{B'(0)}$. We now make the interesting observation that $\rho = 0$ (since $C'(0) = 0$ and $B'(0) = b_1 > 0$) and thus the inequality $r > \rho$ is satisfied for any $r > 0$. Hence, in this case a non-zero level of cooperation will evolve for any positive degree of assortativity.

We obtain the singular strategy $x^\star$ for the game as

$$x^\star = \frac{rb_1}{2rb_2 + 2c_2}. \tag{23}$$

We now choose $x_m = 1$, and for simplicity take $b_1 = 2b_2$. Therefore,

$$x^\star = \frac{1}{1 + \frac{2c_1}{rb_1}}, \tag{24}$$

and $x^\star \in [0, 1]$. Since $rB''(x^\star) - C''(x^\star) = -2b_2 r - 2c_1 < 0$, the singular strategy $x^\star$ is convergent stable, and hence also evolutionarily stable. As a result, an initially monomorphic population in which every individual invests any amount $x_0 \in [0, 1]$, will evolve to a final state in which all individuals cooperate by investing $x^\star$ given by Equation (24). We note in particular that the completely defective initial state $x_0 = 0$ will evolve to the cooperative state given by $x^\star$.

Thus, in this case, cooperation will always evolve from the completely defective initial state $x_0 = 0$ for any non-zero degree of assortativity. Furthermore, since $\frac{\partial x^\star}{\partial r} = \frac{2c_1}{r^2 b_1 (1 + \frac{2c_1}{rb_1})^2} > 0$, the cooperative investment $x^\star$ made in the final state increases with assortativity $r$. Therefore, assortative interactions provide a powerful mechanism for the origin and maintenance of cooperation in the social dilemma defined by the CD game.

### 2.2.3. Continuous Snowdrift Game

Consider again the case of two individuals, John and Bill, who are stuck in a car on their way home because the road is blocked by a snowdrift. Let us now assume that the amount of effort invested by each individual in shoveling to clear the snow can vary. Each individual benefits from the total investment that they both make to clear the snow, however, each individual only bears the cost of their own investment. Such interactions among pairs of individuals in which the investment made by each individuals is beneficial to both, but involves a cost only to the investor, can be described using the continuous snowdrift (CSD) game [26].

The CSD game concerns two individuals, making investments $x, y \in \mathbb{R}_+$, respectively. These investments have the following effects: the payoff of each individual is increased by $B(x + y)$, where $B(z)$ is a function that specifies the benefit to each individual resulting from the total amount of investment made by both participants, and the payoff to the investor, say the $x$-strategist here, is reduced by $C(x)$, where $C(x)$ is a function that specifies the cost to an individual of making a given investment. Therefore, the payoff $\pi(x, y)$ to an $x$-investor interacting with a $y$-investor is given by

$$\pi(x, y) = B(x + y) - C(x). \tag{25}$$

We shall again assume that there is an upper limit $x_m$ on the possible level of investment, that the cost and benefit functions are smooth and monotonically increasing functions satisfying $C(0) = B(0) = 0$, and $B(x) > C(x)$, for $x \in (0, x_c)$, where $0 < x_c < x_m$.

For an assortatively-mixed population of individuals with assortativity $r$, from Equations (18) and (25), we can write the invasion fitness $\phi_x^r(y)$ as

$$\phi_x^r(y) = (1-r)B(x+y) + rB(2y) - C(y) - B(2x) + C(x). \tag{26}$$

The adaptive dynamics of the investment $x$ is governed by

$$\begin{aligned}
\dot{x} &= D(x) \\
&= \left. \frac{\partial \phi_x^r(y)}{\partial y} \right|_{y=x} \\
&= (1+r)B'(2x) - C'(x).
\end{aligned} \tag{27}$$

Concave Cost and Benefit Functions

We now consider the case in which the cost and benefit functions are concave ($C''(x) < 0$ and $B''(z) < 0$). Saturating benefits are clearly realistic [108–110], and decelerating costs are reasonable when the initiation of cooperative acts is more costly than subsequent increases in cooperative investments. Suppose, as an illustrative example, that we take the cost and benefit functions to be quadratic functions, i.e, suppose $C(x) = -c_2 x^2 + c_1 x$ and $B(z) = -b_2 z^2 + b_1 z$, where $c_1, c_2, b_1, b_2, > 0$.

We obtain the singular strategy $x^\star$ for the CSD game as

$$x^\star = \frac{(1+r)b_1 - c_1}{4(1+r)b_2 - 2c_2}. \tag{28}$$

The singular strategy is convergent stable if

$$b_2 > \frac{c_2}{2(1+r)}, \tag{29}$$

and a repeller if the inequality is reversed. The singular strategy is an ESS if

$$b_2 > \frac{c_2}{1+3r}, \tag{30}$$

and an EBP if the inequality is reversed. Similar conditions for the convergence and evolutionary stability of a singular strategy are derived in [101] for a nonlinear public goods game with assortative interactions, in which relatedness is defined as the expected value of a fraction of the group that is identical by descent to the focal individual.

Thus, an initially monomorphic population in which every individual is investing $x_0 \in [0, x_m]$ will evolve to a final state that crucially depends on the coefficients $c_1, c_2, b_1$, and $b_2$ of the cost and benefit functions, and on the assortativity $r$. If Equation (29) is satisfied, then the population initially evolves to a state in which all individuals are investing $x^\star$—the fate of the population thereafter depends on whether or not Equation (30) is satisfied. If it is satisfied, then $x^\star$ is an ESS and the population remains at this end state permanently. Otherwise, $x^\star$ is an EBP and the population splits into two distinct phenotypic clusters, which diverge evolutionarily from each other. If, however, Equation (30) is not satisfied, then $x^\star$ is a repeller, and the fate of the population depends on the initial strategy $x_0$—if $x_0 < x^\star$, then the population evolves to the zero-investment state $x = 0$, while, if $x_0 > x^\star$, then the population evolves to the maximum-investment state $x = x_m$. We note that $\frac{\partial x^\star}{\partial r} = 2(2b_2 c_1 - b_1 c_2) > 0$, if $\frac{c_1}{b_1} > \frac{c_2}{2b_2}$, and thus if the latter inequality holds then increasing assortativity results in higher levels of cooperation $x^\star$. For given cost and benefit functions, both the location and the nature of the singular strategy $x^\star$ varies as the degree of assortative interactions $r$ changes. The varying nature

of the singular strategy $x^\star$ with $r$ in turn results in very different end-states for the evolutionary dynamics. These end-states are: an evolutionary repeller in which all individuals either invest nothing or invest the maximum possible amount, depending on the initial state of the population; an ESS, in which all individuals invest the same amount, given by the singular strategy $x^\star$; or an end-state in which evolutionary branching has occurred, leading to the coexistence of high and low investing individuals. This latter outcome represents a two-fold social dilemma—referred to as the "tragedy of the commune" [26]—not only is the total level of investment socially inefficient, but in addition evolutionary dynamics forces an unequal division in the levels of investment. We observe from Equation (30) that increasing assortativity inhibits evolutionary branching in the CSD game. Thus, assortative interactions have a dual action on the evolutionary dynamics of the CSD game—increased assortativity first leads to higher levels of cooperation and second reduces the potential for evolutionary branching, thereby reducing the likelihood of unequal levels of cooperation.

### 2.2.4. Continuous Tragedy of the Commons Game

Consider the case of two individuals, John and Bill, who jointly use a finite common-pool resource, such as a common fishing ground or shared Internet bandwidth. Each of them benefits from consuming the resource, but the costs incurred are shared among both. Such interactions between pairs of individuals sharing a common-pool resource in which consumption of the resource benefits the consuming individual but is costly to both individuals, can be described using the continuous tragedy of the commons (CTOC) game [27].

The CTOC game involves two individuals, making investment $x, y \in \mathbb{R}_+$, where in this context, the investment means the level of consumption of a limited common-pool resource, and thus cooperative behavior is identified with lower levels of consumption (i.e., lower levels of investment). The investments have the following effects: the payoff of the investor is increased by $B(x)$, where $B(x)$ is a function that specifies the benefit to an individual obtained from consuming a given amount of the resource, and the payoff of each individual is decreased by $C(x + y)$, where $C(z)$ is a function that specifies the cost to both individuals resulting from a given total level of consumption. Therefore, the payoff $\pi(x, y)$ to an $x$-investor interacting with a $y$-investor is given by

$$\pi(x, y) = B(x) - C(x + y). \tag{31}$$

Again we shall assume that there is an upper limit $x_m$ on the possible level of investment, that the cost and benefit functions are smooth and monotonically increasing functions satisfying $C(0) = B(0) = 0$, and $B(x) > C(x)$, for $x \in (0, x_c)$, where $0 < x_c < x_m$. These assumptions simply reflect the fact that the public resource is both finite and valuable to those consuming it.

For an assortatively-mixed population of individuals with assortativity $r$, from Equations (18) and (31), we can write the invasion fitness $\phi_x^r(y)$ as

$$\phi_x^r(y) = B(y) - rC(2y) - (1 - r)C(x + y) - B(x) + C(2x). \tag{32}$$

The adaptive dynamics of the investment $x$ is governed by

$$\begin{aligned}
\dot{x} &= D(x) \\
&= \left. \frac{\partial \phi_x^r(y)}{\partial y} \right|_{y=x} \\
&= B'(x) - (1 + r)C'(2x).
\end{aligned} \tag{33}$$

#### Convex Cost and Sigmoidal Benefit Functions

We now consider the case in which the cost function is convex and the benefit function is sigmoidal. Accelerating costs represent a realistic assumption and are often observed in nature [111].

Benefits are also often accelerating initially and then saturate, resulting in sigmoidal benefit functions [21]. Suppose, as an illustrative example of this type of cost and benefit functions, we take the cost function to be a quadratic function and the benefit function to be a cubic function, i.e, suppose $C(z) = c_1 z^2$ and $B(x) = -b_3 x^3 + b_2 x^2 + b_1 x$, where $c_1, b_1, b_2, b_3 > 0$.

To simplify the analysis, we take $b_2 = 2b_1$ and $c_1 = b_1$. We obtain the singular strategy $x^\star$ for the CTOC game as

$$x^\star = \frac{\sqrt{16r^2 b_1^2 + 12 b_1 b_3} - 4 r b_1}{6 b_3}. \tag{34}$$

The singular strategy is always convergent stable. Moreover, it is an ESS if

$$b_1 < \frac{\sqrt{16r^2 b_1^2 + 12 b_1 b_3}}{2(1-r)}, \tag{35}$$

and an EBP if the inequality is reversed.

Thus, an initially monomorphic population in which every individual is investing $x_0 \in [0, x_m]$ will evolve to an end-state that depends crucially on the coefficients $c_1, b_1$, and $b_3$ of the cost and benefit functions, and on the assortativity $r$. The population first evolves to a state in which all individuals are investing $x^\star$, and the subsequent fate of the population depends on whether or not Equation (35) is satisfied. If it is satisfied, then $x^\star$ is an ESS, and the population remains in this state permanently. Otherwise, $x^\star$ is an EBP and the population splits into two distinct and diverging phenotypic clusters. We note that $\frac{\partial x^\star}{\partial r} = \frac{4b_1\left(\frac{4rb_1}{\sqrt{16r^2 b_1^2 + 12 b_1 b_3}} - 1\right)}{6b_3} < 0$, if $\frac{4rb_1}{\sqrt{16r^2 b_1^2 + 12 b_1 b_3}} < 1$, and thus if the latter inequality holds then increasing assortativity results in higher levels of cooperation, i.e, lower values of $x^\star$. For given cost and benefit functions, the location and the type of the singular strategy $x^\star$ varies as the degree of assortative interactions $r$ varies. Thus, as the degree of assortative interactions $r$ changes, so does the form of the evolutionary dynamics. The end-state of the evolutionary dynamics can be either an ESS in which all individuals consume the same amount $x^\star$ of the common resource, or it can be an end-state in which evolutionary branching has taken place, resulting in the coexistence of high and low consuming individuals. This latter outcome represents a second tragedy of the commons—not only is the resource over consumed to the detriment of all, but evolutionary dynamics forces an unequal division degree of consumption. We observe from Equation (35) that increasing assortativity inhibits evolutionary branching in the CTOC game. Thus, just as in the case of the CSD game, assortative interactions have a double action on the evolutionary dynamics of the CTOC game, with increased assortativity resulting both in higher levels of cooperation and in a reduced likelihood of unequal levels of consumption.

### 2.3. Individual-Based Model

An individual-based model (IBM), or agent-based model [112], provides a natural alternative method of studying the evolutionary dynamics of a population with assortative interactions, which is complementary to the deterministic approaches discussed above.

#### 2.3.1. Discrete Games

We assume here that the individuals in the population play a symmetric $2 \times 2$ game with the payoff matrix given by Equation (4). We consider a population of $n$ individuals, with an initial strategy profile $\mathbf{s}^0 = \{s_1^0, s_2^0, \ldots, s_n^0\}$ at generation 0, where $s_i^0 \in \{C, D\}$, for $1 \le i \le n$. Each generation of the evolutionary dynamics consists of an asynchronous interaction/update round, which involves sampling the population $n$ times with replacement.

Each interaction/update step at generation $t$ is carried out as follows: in the interaction phase we pick uniformly at random two individuals $i$ and $j$ from the population. The two individuals

$i$ and $j$ interact (i.e., play the symmetric $2 \times 2$ game) with individuals $k \neq i$ and $l \neq j$ respectively. The interactions are assortative, i.e., with probability $r$, the individual $k$ (respectively $l$) is of the same strategy type as $i$ (respectively $j$), and with probability $1 - r$, the individual $k$ (respectively $l$) is chosen uniformly at random from the population. The change of individual $i$'s strategy from one generation to the next is determined by the payoff that player $i$ receives from interacting with player $k$ and the payoff that player $j$ receives from interacting with player $l$. Precisely, if $s_i^{t-1}, s_j^{t-1}, s_k^{t-1}$, and $s_l^{t-1}$ denote the strategies of $i, j, k$, and $l$, respectively, in generation $t - 1$, then in generation $t$ the payoff $P_i$ received by individual $i$ is $\pi(s_i^{t-1}, s_k^{t-1})$ and the payoff $P_j$ received by individual $j$ is $\pi(s_j^{t-1}, s_l^{t-1})$, where $\pi$ is the payoff matrix for the game under consideration. In the update phase the probability that the focal individual $i$ will inherit $j$'s strategy, $p_{i \leftarrow j}$, is determined using the *Fermi* update rule as [19]

$$p_{i \leftarrow j} = \frac{1}{1 + e^{-\beta(P_j - P_i)}}, \tag{36}$$

where the parameter $\beta > 0$ is the "selection strength" of the update rule.

We note that the results of the individual-based simulations described in the next section are robust to changes in the update rule. For example, in addition to employing the Fermi update rule given by Equation (36), we have also simulated the individual-based model using the *replicator* update rule, in which the probability $p_{i \leftarrow j}$ that the focal individual $i$ inherits individual $j$'s strategy is given by [19]

$$p_{i \leftarrow j} = \begin{cases} 0 & \text{if } P_i \geq P_j \\ \frac{P_j - P_i}{P_{\max} - P_{\min}} & \text{otherwise}, \end{cases} \tag{37}$$

where $P_{\max} = \max(P_1, P_2, \ldots, P_n)$, and $P_{\min} = \min(P_1, P_2, \ldots, P_n)$. We find that the evolutionary dynamics of the symmetric $2 \times 2$ games that we study is essentially identical irrespective of which of these update rules we employ. The results presented in the next section arise from simulations using the Fermi update rule (Equation (36)).

2.3.2. Continuous Games

An IBM also provides a natural way of studying the evolutionary dynamics of a population playing a pairwise continuous game with assortative interactions.

We again consider a population of $n$ individuals, now with an initial monomorphic strategy profile $\mathbf{x}^0 = \{x_1^0, x_2^0, \ldots, x_n^0\}$ in generation 0, where $x_1^0 = x_2^0 = \cdots = x_n^0 = x_0 \in (0, x_m)$ and $x_m \in \mathbb{R}_+$. Thus in generation $t = 0$, every individual in the population uses the strategy $x_0$. The interactions/update rounds are carried out in the same manner as in the case of discrete games, but with the following additional step: if during the update round at generation $t$, the focal individual $i$ would have inherited the strategy $x_j^{t-1}$ of the individual $j$, then with probability $\mu$ it instead inherits a mutation of this strategy, picked from a normal distribution with mean $x_j^{t-1}$ and standard deviation $\sigma$. If the strategy space is a finite interval $[a, b]$ then the mutations are taken from a truncated normal distribution on $[a, b]$. We note here that changing the precise way in which the random numbers are generated does not effect the outcome of the adaptive dynamics. For example, it was shown in [113] that including random numbers that are uniformly distributed in addition to having random numbers taken from a truncated normal distribution does not significantly change the evolutionary dynamics of the system considered in that paper.

The simulation results presented in the section below are based on the Fermi update rule given by Equation (36). We have also simulated the IBM using the replicator update rule (Equation (37)), and have found the results obtained using both rules to be qualitatively identical.

## 3. Results from Individual-Based Simulations

### 3.1. Discrete Games

In this subsection we present the main results of simulations using the IBM introduced above for the donation, snowdrift, and sculling games. Additional results can be found in the Supplementary Materials document.

### 3.1.1. Donation Game

Figure 4a,b, respectively, show how the analytically predicted and simulated values of $p_\infty$ vary with the assortativity $r \in [0,1]$ and the cost-to-benefit ratio $\rho \in (0,1)$. Figure 4c shows how $p_\infty$ varies with $r$ when $\rho = 0.25$ and Figure 4d shows how $p_\infty$ varies with $\rho$ when $r = 0.25$. The results are in excellent agreement with the analysis given above, showing that a population of individuals playing the donation game will evolve to the all-cooperator state ($p_\infty = 1$) if $r > \rho$ and to the all-defector state ($p_\infty = 0$) if the inequality is reversed.

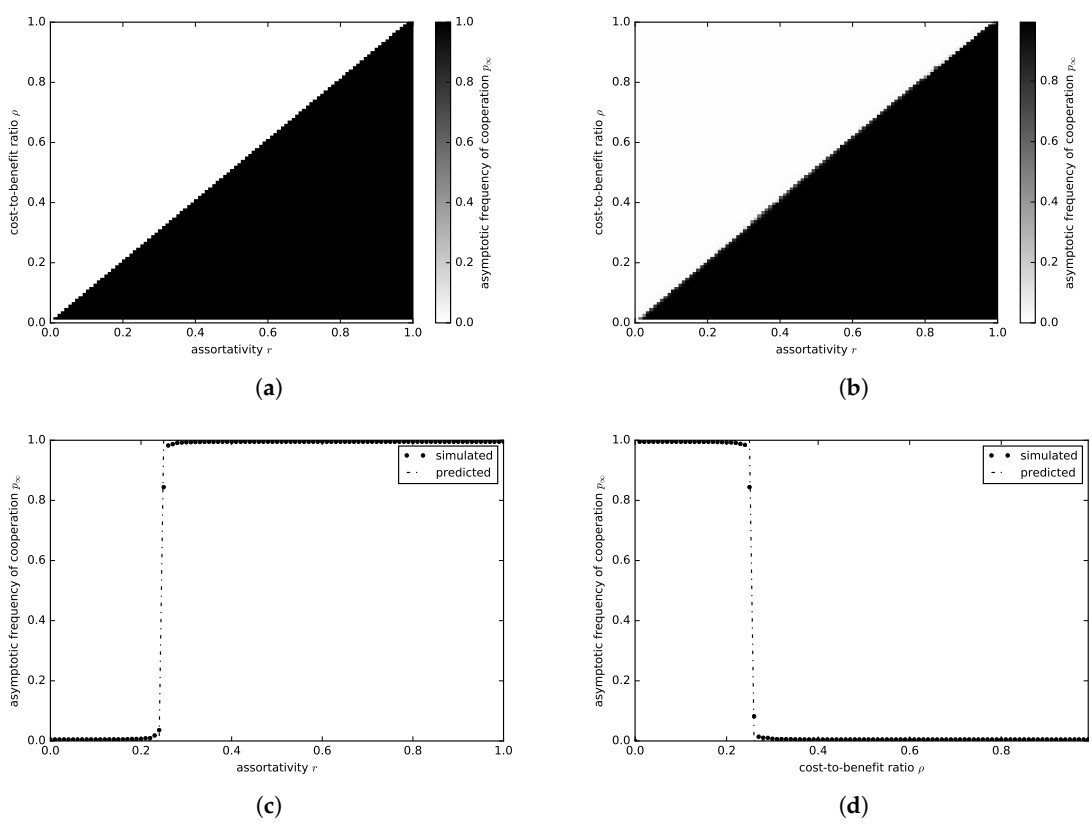

**Figure 4.** Variation of the long-term frequency $p_\infty$ of cooperators with assortativity $r \in [0,1]$ and cost-to-benefit ratio $\rho \in (0,1)$ for the donation game. (**a**) $p_\infty$ (analytically predicted) versus $r$ and $\rho$. (**b**) $p_\infty$ (simulated) versus $r$ and $\rho$. (**c**) $p_\infty$ versus $r$ when $\rho = 0.25$. (**d**) $p_\infty$ versus $\rho$ when $r = 0.25$. Parameters: $n = 10{,}000$, $p_0 = 0.5$, and $\beta = 1$.

### 3.1.2. Snowdrift Game

Figure 5a,b, respectively, show how the analytically predicted and simulated values of $p_\infty$ vary with the assortativity $r \in [0,1]$ and the cost-to-benefit ratio $\rho \in (0,1)$. Figure 5c shows how $p_\infty$ varies with $r$ when $\rho = 0.75$ and Figure 5d shows how $p_\infty$ varies with $\rho$ when $r = 0.25$. The results agree very well with the analysis, which indicates that in a population of individuals playing the snowdrift game, increasing the assortativity $r$ has the effect of increasing the fraction of cooperators at equilibrium, while increasing the cost-to-benefit ratio $\rho$ has the effect of decreasing the fraction of cooperators at

equilibrium. Thus, cooperation is promoted by increasing assortativity $r$, with a transition to complete cooperation occurring when $r > \frac{\rho}{2}$.

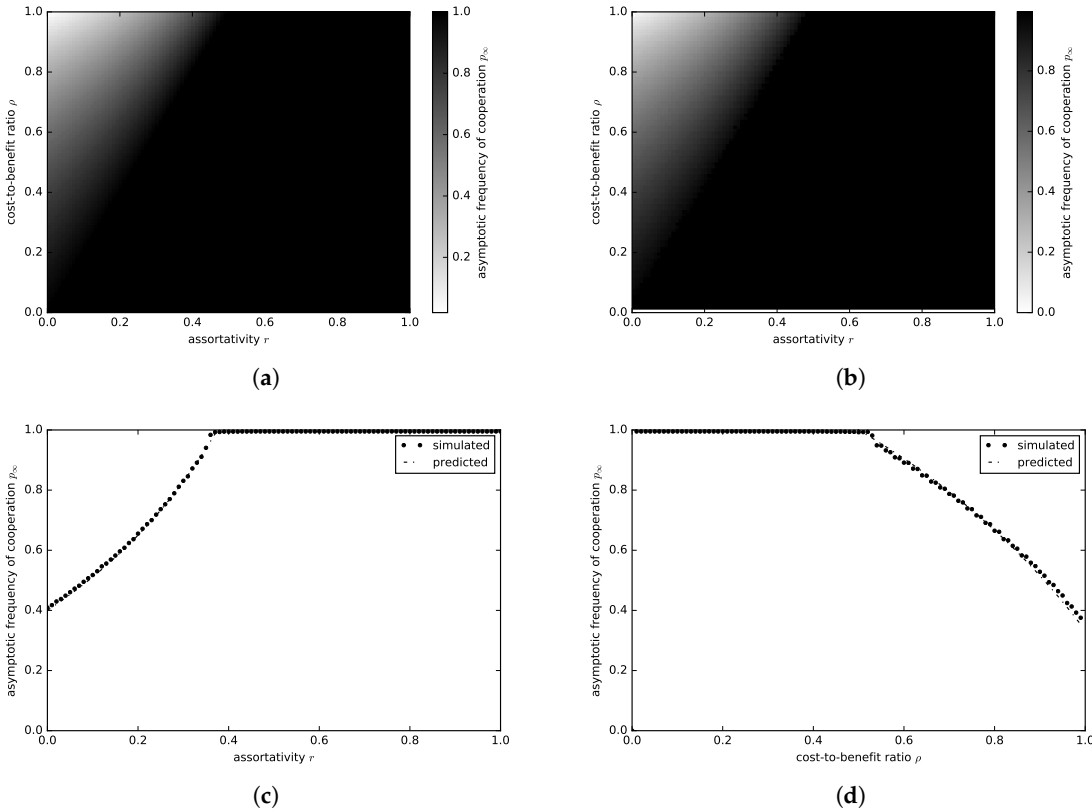

**Figure 5.** Variation of the long-term frequency $p_\infty$ of cooperators with assortativity $r \in [0, 1]$ and cost-to-benefit ratio $\rho \in (0, 1)$ for the snowdrift game. (**a**) $p_\infty$ (analytically predicted) versus $r$ and $\rho$. (**b**) $p_\infty$ (simulated) versus $r$ and $\rho$. (**c**) $p_\infty$ versus $r$ when $\rho = 0.75$. (**d**) $p_\infty$ versus $\rho$ when $r = 0.25$. Parameters: $n = 10{,}000$, $p_0 = 0.5$, and $\beta = 1$.

### 3.1.3. Sculling Game

Figure 6a,b respectively show how the predicted and simulated values of $p_\infty$ vary with assortativity $r \in [0, 1]$ and cost-to-benefit ratio $\rho \in (\frac{1}{2}, \frac{3}{2})$. Figure 6c shows how $p_\infty$ varies with $r$ when $\rho = 1.2$ and Figure 6d shows how $p_\infty$ varies with $\rho$ when $r = 0.25$. The results are in very good agreement with the analysis, which indicates that in a population of individuals playing the sculling game, increasing the assortativity $r$ has the effect of increasing the basin of attraction around $\hat{p} = 1$, thus decreasing the value of the initial fraction $p_0$ of cooperators that is needed for the population to evolve to the all-cooperator state. Contrarily, increasing the cost-to-benefit ratio $\rho$ has the effect of increasing the basin of attraction around $\hat{p} = 0$, thus increasing the value of the initial fraction $p_0$ of cooperators that is needed for the population to evolve to the all-cooperator state. Thus, cooperation is promoted by increasing assortativity $r$, with a transition to complete cooperation occurring when $r > \frac{2\rho}{3} - \frac{1}{3}$.

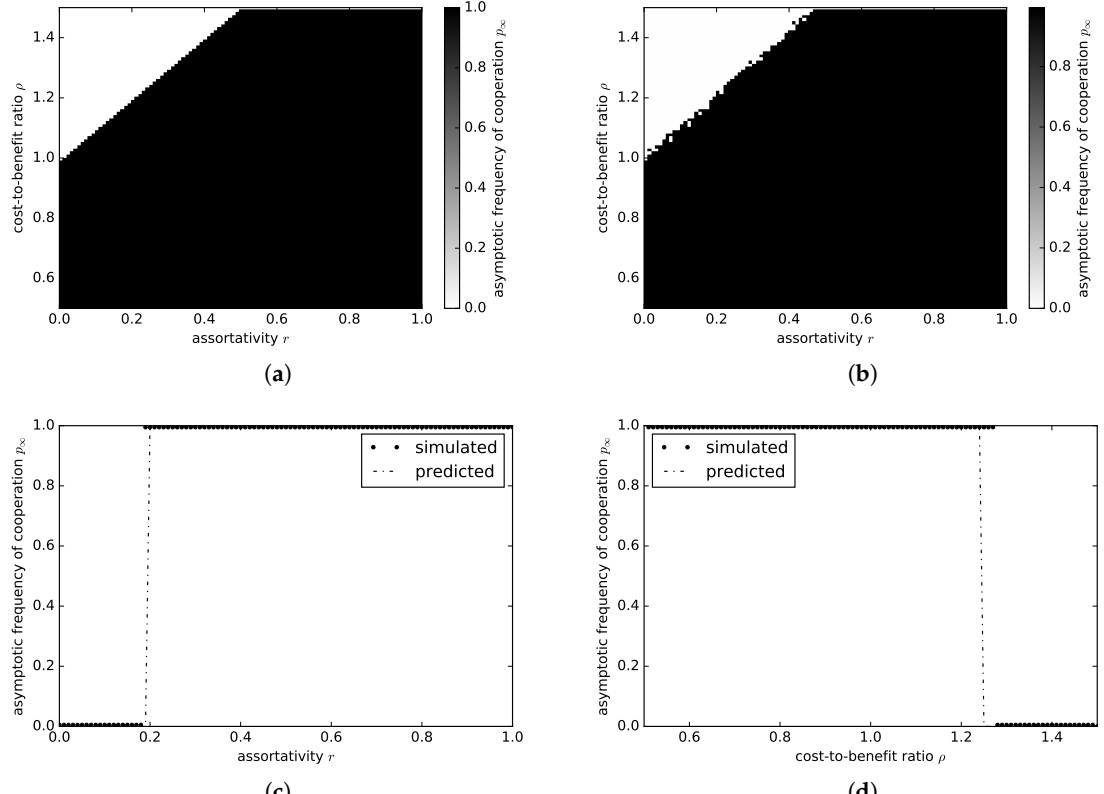

**Figure 6.** Variation of the long-term frequency $p_\infty$ of cooperators with assortativity $r \in [0,1]$ and cost-to-benefit ratio $\rho \in (\frac{1}{2}, \frac{3}{2})$ for the sculling game. (**a**) $p_\infty$ (analytically predicted) versus $r$ and $\rho$. (**b**) $p_\infty$ (simulated) versus $r$ and $\rho$. (**c**) $p_\infty$ versus $r$ when $\rho = 1.2$. (**d**) $p_\infty$ versus $\rho$ when $r = 0.25$. Parameters: $n = 10{,}000$, $p_0 = 0.5$, and $\beta = 1$.

## 3.2. Continuous Games

In this subsection we present the main results of simulations using the IBM introduced above for the continuous donation (CD), continuous snowdrift (CSD), and continuous tragedy of the commons (CTOC) games. Additional results can be found in the Supplementary Materials document.

### 3.2.1. Continuous Donation Game

We first consider the CD game with linear cost and benefit functions $C(x) = cx$ and $B(x) = bx$, where $b > c$. We denote the ratio $\frac{c}{b}$ by $\rho$ and refer to it as the cost-to-benefit ratio. The condition $r > \rho$ that promotes cooperative investments, where $r$ is the degree of assortativity, is similar to the one for the discrete donation game, and hence we study the CD game using a similar set of plots.

Figure 7a,b, respectively, show how the analytically predicted and simulated values of the long-term mean strategy $\bar{x}_\infty$ (taken over the last 10% of the generations) varies with the assortativity $r \in [0,1]$ and the cost-to-benefit ratio $\rho \in (0,1)$. Figure 7c shows how $\bar{x}_\infty$ varies with $r$ when $\rho = 0.26$ and Figure 7d shows how $x_\infty$ varies with $\rho$ when $r = 0.26$. The results are in excellent agreement with the analysis, which shows that in a population of individuals playing the CD game with linear cost and benefit functions, individuals will asymptotically invest the maximum amount $x_m$ if $r > \rho$ and make zero investment if the inequality is reversed.

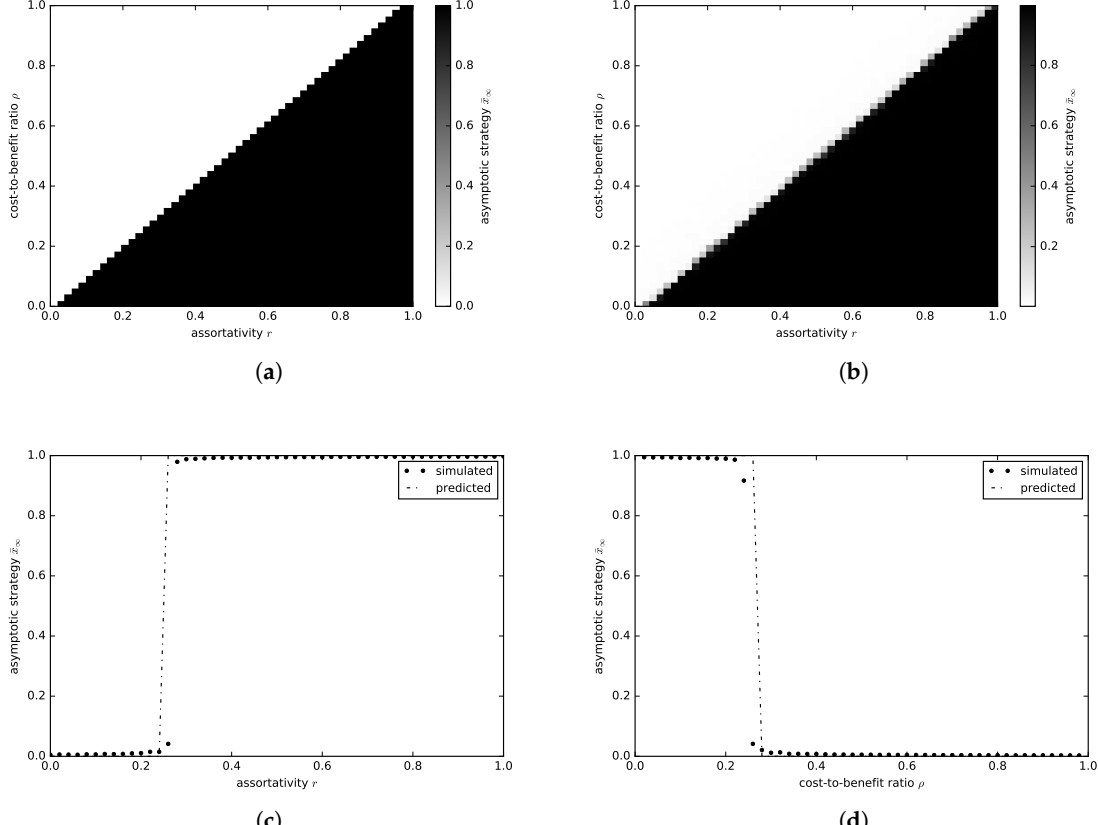

**Figure 7.** Variation of the long-term mean strategy $\bar{x}_\infty$ with assortativity $r \in [0, 1]$ and cost-to-benefit ratio $\rho \in (0, 1)$ in the CD game with linear cost and benefit functions: $C(x) = cx$ and $B(x) = bx$ with $b > c$. (**a**) $\bar{x}_\infty$ (analytically predicted) versus $r$ and $\rho$. (**b**) $\bar{x}_\infty$ (simulated) versus $r$ and $\rho$. (**c**) $\bar{x}_\infty$ versus $r$ when $\rho = 0.26$. (**d**) $\bar{x}_\infty$ versus $\rho$ when $r = 0.26$. Parameters: $n = 10{,}000$, $x_0 = 0.1$, $x_m = 1$, $\mu = 0.01$, $\sigma = 0.005$, and $\beta = 1$.

We next consider the CD game with quadratic cost and benefit functions $C(x) = c_1 x^2$ and $B(x) = -b_2 x^2 + b_1 x$, where $c_1, b_1, b_2 > 0$. We let $b_1 = 2b_2$, in which case the singular strategy, given by Equation (24), is convergent stable and hence also evolutionarily stable.

Figure 8 shows the variation of the distribution of the long-term values $x_\infty$ (taken over the last 10% of the generations) of strategies with assortativity $r$; the dotted line indicates the singular strategy $x^\star$. The results are in very good agreement with the analysis, which indicates that in a population of individuals playing the CD game with quadratic cost and benefit functions, individuals will in the long term invest an amount given by the evolutionarily stable singular strategy $x^\star$, which increases with assortativity $r$.

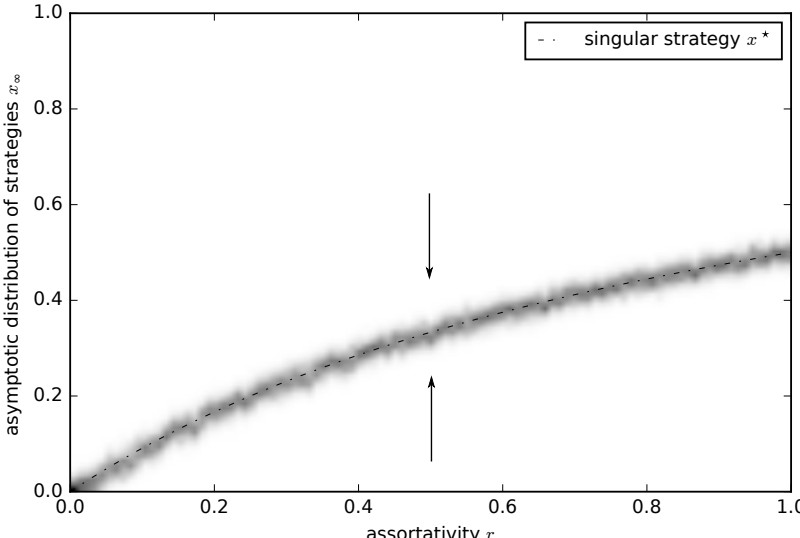

**Figure 8.** Variation of the distribution of long-term strategy values $x_\infty$ with assortativity $r$ in the CD game with quadratic cost and benefit functions: $C(x) = x^2$ and $B(x) = -x^2 + 2x$. Parameters: $x_0 = 0.1$, $x_m = 1$, $\mu = 0.01$, $\sigma = 0.005$, and $\beta = 1$. Arrows indicate the direction of evolutionary change.

### 3.2.2. Continuous Snowdrift Game

The next game we consider is the CSD game with quadratic cost and benefit functions $C(x) = -c_2 x^2 + c_1 x^2$ and $B(x) = -b_2 x^2 + b_1 x$, where $c_1, c_2, b_1, b_1 > 0$. The singular strategy for the game, given by Equation (28), is convergent stable if the inequality given by Equation (29) is satisfied and a repeller if the inequality is reversed. On the other hand, the singular strategy is an ESS if the inequality given by Equation (30) is satisfied and an EBP otherwise.

Figure 9a,b show the variation of the distribution of the asymptotic strategy values $x_\infty$ (taken over the last 10% of the generations) with assortativity $r$. The dotted line indicates the singular strategy $x^\star$. The value of $r$ in (a) at which the singular strategy transitions from an EBP to an ESS is determined by equating the two sides of Equation (30) and solving for $r$. For the given choice of parameters, the transition point (indicated by the dashed vertical line) is $r = 0.2$. Similarly, the value of $r$ in (b) where the singular strategy changes from a repeller where every individual makes zero asymptotic investment to one in which all individuals make the maximum asymptotic investment $x_m$ can be obtained from Equation (29). For the given choice of parameters, the transition point (indicated by the dashed vertical line) is $r = 0.13$. The results are in very good agreement with the analysis, which shows that in a population of individuals playing the CSD game with quadratic cost and benefit functions, for suitable values of the coefficients, the singular strategy can change from an EBP to an ESS as the assortativity increases. Similar results are presented in [101] for a nonlinear public goods game with assortative interactions, in which evolutionary attractors increase with relatedness (defined as the expected value of a fraction of the group that is identical by descent to the focal individual) while evolutionary repellers decrease with relatedness.

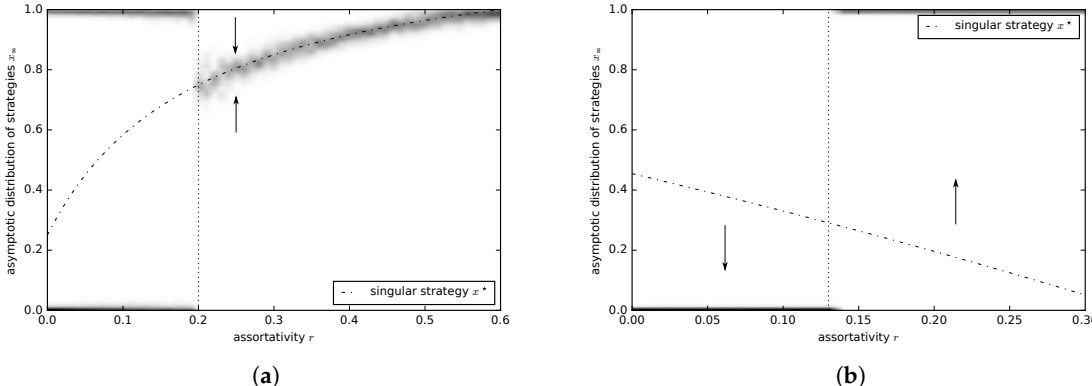

**Figure 9.** Variation of the distribution of asymptotic strategy values $x_\infty$ with assortativity $r$, in a CSD game with quadratic cost function and quadratic benefit function. (**a**) $C(x) = -1.6x^2 + 4.8x$ and $B(x) = -x^2 + 5x$, and (**b**) $C(x) = -1.5x^2 + 4x$ and $B(x) = -0.2x^2 + 3x$. Parameters: $n = 10{,}000$, $x_0 = 0.3$, $x_m = 1$, $\mu = 0.01$, $\sigma = 0.005$, and $\beta = 1$. Arrows indicate the direction of evolutionary change.

### 3.2.3. Continuous Tragedy of the Commons Game

Finally, we consider the CTOC game with quadratic cost and cubic benefit functions $C(x) = c_1 x^2$ and $B(x) = -b_3 x^3 + b_2 x^2 + b_1 x$. If we let $b_2 = 2b_1$ and $c_1 = b_1$, the singular strategy for the game is given by Equation (34), and is an ESS if the inequality given by Equation (35) is satisfied and an EBP otherwise.

Figure 10 shows the variation of the distribution of asymptotic strategy values $x_\infty$ (taken over the last 10% of the generations) with assortativity $r$. The dotted line indicates the singular strategy $x^\star$. The value of $r$ at which the singular strategy transitions from an EBP to an ESS is obtained by equating the two sides of Equation (35) and solving for $r$. For the given choice of parameters, the transition point (indicated by the dashed vertical line) is $r = 0.3$. The results are in excellent agreement with the analysis, which show that in a population of individuals playing the CTOC game with quadratic cost and cubic benefit functions, for suitable values of the coefficients, the singular strategy can transition from an EBP to an ESS with increasing assortativity.

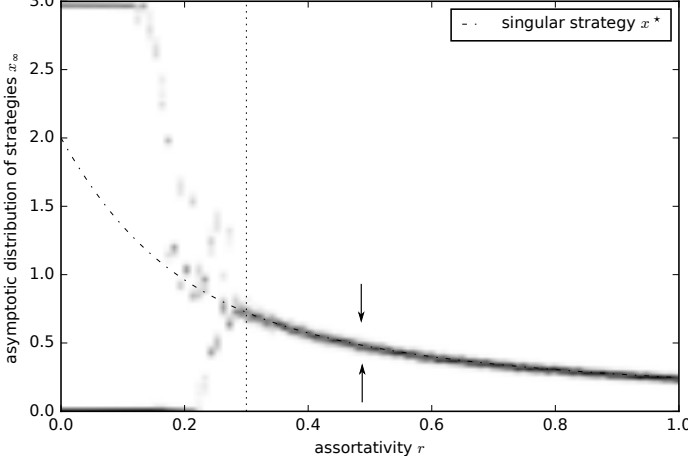

**Figure 10.** Variation of the distribution of asymptotic strategy values $x_\infty$ with assortativity $r$, in a continuous tragedy of the commons (CTOC) game with quadratic cost and cubic benefit functions: $C(x) = x^2$ and $B(x) = -0.0834x^3 + 2x^2 + x$. Parameters: $n = 10{,}000$, $x_0 = 0.1$, $x_m = 3$, $\mu = 0.01$, $\sigma = 0.005$, and $\beta = 1$. Arrows indicate the direction of evolutionary change.

## 4. Discussion

In this work we have undertaken a detailed and systematic investigation of the evolution of cooperation in a wide variety of both discrete strategy and continuous strategy social dilemmas with assortative interactions modeled in the manner proposed by [79,80]. For the discrete strategy social dilemmas that we have studied—the donation, snowdrift and sculling games—we find that in all cases the frequency of cooperation increases with increasing assortativity. In these social dilemmas the transition to complete cooperation is governed by Hamilton's rule for the donation game, and natural analogs of it for the snowdrift and sculling games. Thus, the entire population evolves to cooperators in the donation game if $r > \rho$, in the snowdrift game if $r > \frac{\rho}{2}$, and in the sculling game if $r > \frac{\rho - \frac{1}{2} - p_0}{\frac{3}{2} - p_0}$ (where $p_0$ is the initial frequency of cooperators). The results for the donation game are essentially well-known [13,79,80,89–98], while the results for the snowdrift and sculling games are more novel. Interesting related work is [98], which studies the effect of relatedness on the maintenance of cooperation in prisoner's dilemma, hawk-dove, and stag hunt games. The definition of assortativity and of the games considered in [98] are formulated differently from those adopted here, and we hope that our results may be complementary to those of [98].

These results apply quite directly to games between relatives due to the fact that Grafen's geometric interpretation of relatedness allows the degree of assortativity $r$ to be identified with the degree of relatedness between the interacting individuals [79,80]. Thus, our results indicate that the emergence of complete cooperation in populations of related individuals playing the snowdrift and sculling games should be governed by the variants of Hamilton's rule described above.

Our results also elucidate the evolution of cooperation in discrete strategy social dilemmas on networks, since evolutionary dynamics in structured populations can result in assortative interactions between the individuals. For example, consider the results obtained for the evolution of cooperation in the donation, snowdrift and sculling games on networks in [19]. Comparing the condition found in [19] for complete cooperation to occur in the donation game on networks of mean degree $k$ (namely, $\frac{1}{\rho} > k - 1$) with the condition obtained here for assortative interactions ($r > \rho$) suggests that the effective degree of assortativity resulting from network interactions in the framework studied in [19] is $r = \frac{1}{k-1}$. Now taking this value of $r$ and inserting it into the condition obtained here for the evolution of complete cooperation in the snowdrift game ($r > \frac{\rho}{2}$) then gives that the transition to complete cooperation in the snowdrift game on a network of mean degree $k$ should be governed by the rule that complete cooperation prevails if $\frac{1}{\rho} > \frac{k-1}{2}$. This latter rule is consistent with the results found in [19]. Similarly, taking the value $r = \frac{1}{k-1}$ in the condition derived here for the evolution of cooperation in the sculling game ($r > \frac{\rho - \frac{1}{2} - p_0}{\frac{3}{2} - p_0}$, and taking the initial frequency of cooperators to be $p_0 = \frac{1}{2}$ as was the case in [19]) gives the condition for cooperation to evolve in the sculling game on networks found in [19]: namely, that cooperation prevails if $\frac{1}{\rho - 1} > k - 1$. Thus we find that despite the simple manner in which assortativity has been modeled here, we obtain results which are applicable to much more subtle systems in which assortativity emerges as a complex, self-organized, phenomenon.

For the continuous strategy social dilemmas that we have considered—the continuous donation, continuous snowdrift and continuous tragedy of the commons games—we also find that the level of cooperation increases with increasing assortativity. An additional interesting finding, and perhaps the most significant result that we have obtained, is that the propensity for evolutionary branching to occur in the continuous snowdrift and continuous tragedy of the commons games is reduced as the degree of assortative interactions increases. Thus, assortativity plays a doubly beneficial role in the evolution of cooperation in these games both by increasing the overall level of cooperation and by reducing the likelihood of unequal and unfair outcomes. We conjecture that this inhibiting effect of assortative interactions on evolutionary branching is a general phenomenon that holds for a broader class of evolutionary systems than merely the continuous strategy games that we have considered here. The interesting work [101] has also explored similar topics to those studied in this paper in that it

considers the effect of relatedness on cooperation in a multi-player version of the continuous snowdrift game. In [101] relatedness is introduced in a different, and more general, manner to our definition of assortativity, through the consideration of a probability distribution over the number of co-players that are identical-by-descent to a focal individual. Thus, the results obtained in [101] for the evolutionary dynamics of the multi-player continuous snowdrift game include the results we have obtained for the two-player continuous snowdrift game as a special case (the precise connection between the two approaches is that if we take $N = 2$, $\Pr(1) = 1 - r$ and $\Pr(2) = r$ in Equation (3) for the invasion fitness in [101] then we obtain an expression that is equivalent to our Equation (26) for the invasion fitness). Of particular interest is the result found in [101] that increased relatedness reduces the possibility of evolutionary branching in the continuous multi-player snowdrift game. Thus, our result that the propensity for evolutionary branching to occur in the continuous snowdrift games decreases as the degree of assortative interactions increases is a special case of the findings of [101]. Since our formalism for studying the effect of assortativity on the evolution of cooperation in continuous strategy games is different from that of [101] we hope that these distinct methodologies may prove to be complementary. It would also be most interesting to apply the methods of [101] to study the evolutionary dynamics of the continuous donation and continuous tragedy of the commons games.

Our results for evolutionary branching in the continuous snowdrift and continuous tragedy of the commons games with assortative interactions imply corresponding results for evolutionary branching in these games played between relatives. We expect, therefore, that evolutionary branching in these games will be inhibited as the degree of relatedness between the interacting individuals increases. Again, it is plausible to conjecture that this phenomenon is not restricted only to those games that we have considered here but holds more widely.

Furthermore, we observe that our results on evolutionary branching in the continuous snowdrift and continuous tragedy of the commons games with assortative interactions may have interesting implications for the evolutionary dynamics of these games on networks. Since one would expect the assortativity produced by interactions on a network to increase as the mean degree of the network decreases, it is therefore natural to conjecture that evolutionary branching in the continuous snowdrift and continuous tragedy of the commons games on networks will be inhibited as the mean degree of the network decreases. We have studied the evolutionary dynamics of the continuous snowdrift and continuous tragedy of the commons games on networks through simulations, and have found that evolutionary branching is inhibited on networks of low mean degree, exactly as predicted by our results (a detailed discussion of the effects of network structure on evolutionary branching in these games will be given elsewhere). Again, it is natural to conjecture that such an increase in the inhibition of evolutionary branching on networks of low mean degree may be a general phenomenon applicable to many evolutionary systems.

**Supplementary Materials:** The following are available at http://www.mdpi.com/2073-4336/11/4/41/s1.

**Author Contributions:** Conceptualization, T.K.; formal analysis, S.I. and T.K.; software, S.I.; visualization, S.I.; writing—original draft, S.I. and T.K.; writing—review and editing, S.I. and T.K. All authors have read and agreed to the published version of the manuscript.

**Funding:** This research received no external funding.

**Conflicts of Interest:** The authors declare no conflict of interest.

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
