# Peer review of "Evolution of Cooperation in Social Dilemmas with Assortative Interactions"

_games, doi:10.3390/g11040041_

Round 1
Reviewer 1 Report
In the submitted manuscript, the authors systematically investigate the consequences of positive assortment – modeled as a prescribed probability of interacting with an individual of one’s own type. Their investigation includes both discrete strategy games and continuous strategy games. For each respective game, the authors find conditions for the evolution of cooperation that are formally similar to Hamilton’s rule. For games with continuous strategies, they show that evolutionary branching is inhibited by positive assortment. Their results are corroborated by individual-based simulations that show good agreement with the analytical predictions.
I believe this submission has value as an overview of how positive assortment affect the evolutionary dynamics for a broad class of games. With this said, the results are exactly as anticipated – increased assortment increases the propensity for evolution to evolve. The possibly most interesting result, as the authors themselves note, is that positive assortment inhibits evolutionary branching in continuous strategy games. This result, however, is not novel. The evolution of cooperation in continuous strategy games with positive assortment have been studied by Coder & Brännström (2018). They also found that positive assortment (“relatedness” in the words of these authors) reduces the propensity for evolutionary branching. Their analysis of the continuous snowdrift game appears to be more general – allowing for any number of interacting players, a broad range of benefit and cost functions, and giving conditions for when coexistence of a protected dimorphism of full cooperators and full defectors are possible. I recommend the authors to better convey what is novel with their contribution and better acknowledge previous work, in particular Coder & Brännström (2018). Perhaps the authors’ analysis of the continuous strategy games can also be expanded by building on this prior work.
The manuscript is generally well written, but I find it unnecessarily long and think it will have a higher chance of being read if it can be shortened to better bring out the key elements. This observation applies generally, but I particularly think it would be good to shorten and streamline the introduction (lines 99-127 can, for example, easily be moved to the discussion or cut entirely) as well as to move most of the current “Results” section – which is really only a corroboration of analytical results – to the supplementary material. The real results of this manuscript are the analytical findings and they deserve to be presented as such, possibly briefly complemented with individual-based results in the main text to corroborate key findings or illustrate phenomena not covered by analytical results. I also recommend the authors to add figures to illustrate the dynamics of continuous strategy games.
Minor comments:
The line numbers are missing in parts of the document.
L4, and possibly elsewhere. The authors seem to equate cooperation with social dilemmas. In fact, many acts of cooperation (in the dictionary sense of working together) may have a direct benefit to the individual that exceed the costs. This may, for example, be the case with schooling in fish. See Cornforth et al. (2012) for a discussion of how this may occur when benefits are synergetic.
L12. I think the authors are unjustified in almost dismissing mechanisms of cooperation as mere “surface difference”. To me, the key challenge in the evolution of cooperation is to understand how different mechanisms can give rise to positive assortment.
L210. Perhaps emphasize that the definition applies to positive assortative interactions only.
L222-223. It is unclear to me why the authors claim that the payoffs represent a change in the expected number of offspring. First, this leaves out the most important case of imitation dynamics in social evolution, which most of the authors’ examples center around. Second, how does one really interpret a negative payoff of, say, -6 as a change in the expected number of offspring?
P6, e.g. L230-236. To me, the “r-replicator equation” appears to be exactly the replicator equations with different expressions for the payoffs. I recommend that the authors simply state that they use the standard replicator equation and avoid giving the impression that they have derived a novel “r-replicator equation”.
L275. In what sense is positive assortment “a direct mechanism” for promoting cooperation? To me, a mechanism would be a way to achieve positive assortative interactions.
Figures 1-3. The authors may consider adding arrows in panel b to show the direction of evolutionary change. I also think it is very unfortunate that no corresponding figures are shown for continuous strategy games. Why is that? This would be most valuable and helpful.
L286: The authors talk about a “mixed strategy” here (and in other places), but the model description in section 2.1.1 is only about individuals that apply pure strategies. The frequency of individuals that apply a given pure strategy will then change, but in no instance will there be a single individual applying a mixed strategy.
L406. It is not obvious to me that there will always be a single, unique analogue to Hamilton’s rule, nor how it is defined by the authors if this is the case. Here, the authors state that if r>rho, cooperation will take off from the non-cooperative state and be maintained at a non-trivial state. In the discrete snow drift game, Figure 2, this is the case if r>0, at least in the figure shown. For the equivalent relation to Hamilton’s rule, r>rho/2, the authors instead require that cooperation should take off from the non-cooperative state and reach full cooperation. What is required, in terms of the outcome, for a given relation to be “the analogue” to Hamilton’s rule (rather than an analogue)?
Figures 4-17. As mentioned before, I think most of these figure aren’t needed in the main text and could easily be moved to the supplementary. One or a few of the more interesting ones could perhaps be kept if the authors find this useful. I also think the quality of the figures need to improve substantially if they are to be kept in the main text. In particular, the font size needs to increase and I recommend that they authors use grey color (or a non-saturated color) instead of black for pairwise invadability plots (PIP:s).
Cited references:
Coder, K. G., & Brännström, Å. (2018). Effects of Relatedness on the Evolution of Cooperation in Nonlinear Public Goods Games. Games 9, 87.
Cornforth, D.M.; Sumpter, D.J.T.; Brown, S.P.; Brännström, Å. (2012). Synergy and group size in microbial cooperation. Am. Nat. 180, 296–305.
Reviewer 2 Report
Please see the attached document

Reviewer 3 Report
The authors consider the role of assortative interactions for the evolution of cooperation in both discrete-strategy and continuous-strategy social dilemmas. Using the frameworks of replicator equations and adaptive dynamics, they characterize the requisite levels of assortment required to achieve long-time cooperation or non-minimal levels of cooperative effort. In particular, in the continuous-strategy case, they show that assortment can help promote the level of effort at the evolutionarily singular strategy and decrease the parameter regime in which evolutionary branching occurs for both continuous snowdrift games and a tragedy of the commons. This means that assortment can help to solve continuous-strategy social dilemmas both by raising the average level of cooperation in the population and by helping to eliminate unequal divisions of effort for the cooperator and defector strategies that can emerge via evolutionary branching. For both the discrete and continuous cases, the authors relate the level of assortment needed for the promotion of cooperation to inequalities reminiscent of Hamilton's rule.
I think the results for the continuous-strategy games are of particular applied interest, as they show how assortment can help to resolve social dilemmas that arise, for example, in the management of common-pool resources like fisheries. However, below are several comments that I have I believe can help to improve the manuscript and to highlight the interesting results related to evolutionary branching.
- The analysis of the discrete-strategy games via the r-replicator dynamics has also been explored by van Veelan et al (2017), which was referenced in the current manuscript only in the context of its consideration of assortment in continuous-strategy games. In that paper, the authors describe the minimal level of assortment required to make cooperation globally stable under the r-replicator dynamics. These threshold conditions are explored for all Prisoners' Dilemma, Hawk-Dove, and Stag Hunt games, and are given in terms of the payoff matrix entries R, S, T, and P. In the current manuscript, the analysis is done for the donation, snowdrift, and sculling games in terms of a critical cost-to-benefit ratio, which formulates the conditions in a slightly different manner. I believe it is worth mentioning the use of the r-replicator dynamics in the van Veelen et al paper, and this citation can also help to streamline some of the exposition (as the authors can choose, for example, to omit the derivation of the r-replicator dynamics and to shorter the calculations by citing the thresholds from the previous paper).
- The question of assortment in continuous-strategy versions of the Prisoners' Dilemma has also been explored by Brännström and couathors, including in a recent paper published in Games. In Cornforth et al (2012) and Coder Gylling and Brännström (2018), the authors respectively study the role of genetic relatedness in promoting cooperation in continuous-strategy versions of the donation game. The model of relatedness is different in these papers than in the current manuscript, as the authors instead consider a probability distribution over the number of co-players that are identical-by-descent to a focal individual, and then they quantify this assortment distribution based upon the mean and variance of assortative interactions. As increasing of assortment in those models tended to suppress evolutionary branching, it may be interesting to consider how this version of assortment could impact the possible branching in the continuous-strategy snowdrift game and tragedy of the commons.
- The paper is also fairly long and has several sections taking different methodological approaches. It may help the readability of the paper to make certain portions of the paper more concise. For example, after having introduced the the expression for and assumptions regarding invasion fitness for any continuous-strategy game, the authors include a paragraph re-describing this idea before introducing the particular expression for the donation game and snowdrift game (before Equations 29 and 37, respectively). I feel that decreasing this repetition will help to highlight the interesting results from the adaptive dynamics section.
Coder Gylling, K., & Brännström, Å. (2018). Effects of relatedness on the evolution of cooperation in nonlinear public goods games. Games, 9(4), 87.
Cornforth, D. M., Sumpter, D. J., Brown, S. P., & Brännström, Å. (2012). Synergy and group size in microbial cooperation. The American Naturalist, 180(3), 296-305.
van Veelen, M., Allen, B., Hoffman, M., Simon, B., & Veller, C. (2017). Hamilton's rule. Journal of theoretical biology, 414, 176-230.
Reviewer 4 Report
This paper intensively investigated the effects of assortative interaction on various classes of games on cooperation and found overall positive effects on the evolution of cooperation. I can easily imagine that skilled mathematical biologists think it is obvious that assortative interaction promotes cooperation in various settings. However, most of the researchers with a weak background in mathematical biology still tend to interpret Hamilton's r as a relatedness between two individuals. It is of significant importance to spread the interpreting it as a degree of assortative interaction. I believe the current paper contributes such a view by carefully demonstrating the positive roles of assortative interaction on the evolution of cooperation.
It might be informative for readers if it is mentioned the importance of assortative interaction on the evolution of cooperation is also derived from the Price equation as is discussed by Henrich (2004, Journal of Economic Behavioral and Organization) and other researchers so that it becomes evident that assortative interaction is not just one of the causal mechanisms of cooperation but it is a general feature of mechanism promoting the evolution of cooperation, the point discussed by the authors inlines 56-77.
Author Response
We thank the reviewer for the positive feedback on our work.
Round 2
Reviewer 1 Report
Having read the revised manuscript, I feel that it has considerably improved but that two key issues still need to be addressed before the manuscript can be accepted for publication:
1. I appreciate that the authors now reference Coder Gylling & Brännström (2018) and the prior work by Cornforth et al. (2012), but in several places they misrepresent this work as being “different” or even “quite different” from the approach taken in their paper. However, the analysis of the snowdrift game by Coder Gylling & Brännström is not different, it is only more general. It encompasses the authors analysis obtained as the special case when the group size is two. To see this, consider Eq. (3) for the invasion fitness in Coder Gylling & Brännström and let N=2, Pr(1)=1-r, and Pr(2)=r. One then obtains the invasion fitness
(1-r)*B(r+m)/2 + r*B(2m)/2 – C(m) – B(2r)/2 + C(r)
This is identical to Eq- 21 of the submitted manuscript after setting r=x, m=y and scaling the benefit function by a factor of two. Hence, all results from Coder Gylling & Brännström apply, including their finding that positive assortment reduces the propensity for evolutionary branching. As the analysis by Coder Gylling & Brännström is more general, they state results in terms of mean and variance. For the Bernoulli distribution that occurs in the two-player case, they are not independent and both the mean and the variance can be expressed in the single parameter r, specifically mean: r, variance: r(1-r). Hence, all their results for the two-player case can be cast in this single paramter. The mean and variance can only be independently varied (to some extent) for three players or more.
There is nothing wrong in studying the important special case of two players, but the authors should state clearly that the analysis of the snowdrift game by Coder Gylling & Brännström is more general, not claim that it is different. Also, in the passage in the discussion starting on L727, the authors should clearly acknowledge that Coder Gylling & Brännström have previously found that evolutionary branching is impeded by assortment. This is not a new finding by the authors – their contribution is limited to corroborating/verifying an already published finding.
I also reiterate my suggestion from last time, that the authors could extend their analysis using the techniques developed by Coder Gylling & Brännström. In particular, it would be great to see when a protected dimorphism is possible in the three continuous strategy games considered.
2. In my previous referee report I had strongly advised the authors to include figures that illustrate the dynamics in the three continuous strategy games considered, similar to the figures for the discrete strategy games. Specifically, I wrote:
“I … think it is very unfortunate that no corresponding figures are shown for continuous strategy games. Why is that? This would be most valuable and helpful”
The authors did not respond to this suggestion. I think such figures would be immensely helpful to readers and increase the chance the work is read and cited. As the authors’ work is not particularly novel and that the findings are either expected or have previously been published by others, I think they should try to make their results as accessible as possible. A key element is to illustrate the results with figures, also in the continuous-strategy case.
Minor comments:
L15: ”the analouges” –> ”analogues”.
L147 & L787: ”differet manner” -> ”more general manner”
L786: The statement here that the analysis by Coder Gylling & Brännström is ”quite different” is not true. As stated above, it is more general and contain the authors result as a special case for two players.
L772-776: The authors should avoid giving an impression of novelty here. Coder Gylling & Brännström have already published the finding that assortment impedes evolutionary branching in the continuous snowdrift game.
Figure 4-9: The font size of these figures should be increased for accessibility. Consider using titles for the different columns in Figs. 4-6 to identify which column is the IBM and which is analytical.
The arrows in Figure 9a are a bit misleading. In all other figures, the arrows indicate the direction a monomorphic population with that trait value would evolve. Here, the arrows on the left of 0.2 means something different (a monomorphic population would evolve towards the evolutionary branching point). I suggest to either reverse the direction of the arrows and clearly explain their meaning in the caption, or delete the left pair of arrows.
The section name “Results” gives the impression that there are no results in the other sections. I think this is misleading and suggest another name, e.g. “Results from individual-based simulations”
I think the authors have done a good job with shortening the manuscript. Maybe it can still be further shortened? If so, it would probably increase the chances that it will be read and cited.
Author Response
Referee #1's second report argues persuasively that the continuous
snowdrift result can be regarded as a special case of results in the 2018
Games paper by CG&B. I appreciate that the assumptions are sufficiently
different that that is not obvious. Nevertheless, to the extent that it is true,
it must be acknowledged and briefly explained.
The reviewer’s comments were very helpful and we have now fully
acknowledged that our results for the continuous snowdrift game are a special
case of those obtained in CG&B 2018 and have briefly explained the connection
on L800-810.
The Referee also notes a connection of assortativity and the classic
Price equation. This also deserves a mention in the revision, or a
convincing explanation in the response letter on why there actually is no
useful connection.
This is an interesting comment and we have now mentioned this on L105-108.
Reviewer 2 Report
My previous report must have been unclear because the authors failed to do what I asked them to do. For this, I apologize. When I asked for "major revisions" I meant that the paper needed a substantial amount of work to become publishable. Something that required more than a few days and a couple of footnotes.
To begin with, the authors should have reconsidered their position about assortativity being the only mechanism for the emergence of cooperation. In their reply to me, the authors state that "any proximate mechanisms for maintaining cooperation in the prisoner’s dilemma must result in assortative
interactions. This is simply a logical necessity, since in the prisoner’s
dilemma if interactions occur at random then defection always triumphs." I think this is incorrect. If the PD is played repeatedly with a sufficiently long time horizon, then cooperation may emerge even without assortativity (although assortativity facilitates the process.) Again, the authors should read carefully the references I provided in my report.
Second, in their reply to me, the authors say that "we should have made it clear that we were not claiming any novelty for the results on the prisoner’s dilemma — this case is discussed essentially for completeness, while the more novel cases are, for discrete games, the snowdrift game and the sculling game". But this is incorrect because if they read one of the Bergstorm papers that I mentioned in my report, they will find a discussion of the stug-hunt game, which is essentially their snowdrift game.
The authors should realize that if a referee finds questionable statements virtually in any part of the paper she can easily check, then she is prone to distrust all the other parts it would take her time to check. Consider that I was given three days (!) to check the current version. How could I possibly ascertain whether some of the results on continuous games are not already somewhere in the literature?
My suggestion to the authors is to read more carefully the literature. Avoid making sweeping statements about assortativity as the only ultimate mechanism for the emergence of cooperation. Avoid reproducing results that have already been proved in the literature. Focus on the new results they have, assuming of course that they add something to what we already know on this topic.
The paper should be much shorter and contain much less material.
Author Response
Referee 2 is not persuaded that TfT (and other repeated game strategies
that support cooperation in repeated games) is usefully thought of as an
instance of assortativity.
I think that skepticism is well taken, at least for the way assortativity is
formulated in the present paper. I think the somewhat facile introductory
remarks on TfT etc need to be modified or dropped.
We agree that these remarks may not be helpful and we have now removed
them.